# (Mis)Fitting: A Survey of Scaling Laws

**Margaret Li\*, Sneha Kudugunta\*, Luke Zettlemoyer**
{margsli,snehark}@cs.washington.edu

## Abstract

Modern foundation models rely heavily on using scaling laws to guide crucial training decisions. Researchers often extrapolate the optimal architecture and hyper parameters settings from smaller training runs by describing the relationship between, loss, or task performance, and scale. All components of this process vary, from the specific equation being fit, to the training setup, to the optimization method. Each of these factors may affect the fitted law, and therefore, the conclusions of a given study. We discuss discrepancies in the conclusions that several prior works reach, on questions such as the optimal token to parameter ratio. We augment this discussion with our own analysis of the critical impact that changes in specific details may effect in a scaling study, and the resulting altered conclusions. Additionally, we survey over 50 papers that study scaling trends: while 45 of these papers quantify these trends using a power law, most under-report crucial details needed to reproduce their findings. To mitigate this, we we propose a checklist for authors to consider while contributing to scaling law research.

## 1 Introduction

Training at the scale seen in recent large foundation models (Dubey et al., 2024; OpenAI, 2023; Reid et al., 2024) is an expensive and uncertain process. Given the infeasibility of hyperparameter tuning multi-billion parameter models, researchers extrapolate the optimal training setup from smaller training runs. More precisely, scaling laws (Kaplan et al., 2020) are used to study many different aspects of model scaling. Scaling laws can guide targets for increasing dataset size and model size in pursuit of desired accuracy and latency for a specific deployment scenario, study architectural improvements, determine optimal hyperparameters and assist in model debugging.

Scaling laws are often characterized as power laws between the loss and size of the model and dataset, and are seen in several variations (Section 2). These laws are found empirically by training models across a few orders of magnitude in model size and dataset size, and fitting the loss of these models to a proposed scaling law. Each component of this process varies in the reported literature, from the specific equation being fit, to the training setup, and the optimization method, as well as specific details for selecting checkpoints, counting parameters and the objective loss optimized during fitting.

Table 1: We provide a summary of the papers surveyed, highlighting the reproducibility challenges endemic to scaling law papers.

| # Papers... | |
|---|---|
| Surveyed | 51 |
| With a quantified scaling law | 45 |
| With a description of training setup | 40 |
| With a definition of equation variables | 36 |
| With a description of evaluation | 29 |
| With a description of curve fitting | 28 |
| With analysis code | 19 |
| With metric scores or checkpoints provided | 17 |

Changes to this setup can lead to significant changes to the results, and therefore completely different conclusion to the study. For example, Kaplan et al. (2020) studied the optimal allocation of compute budget, and found that dataset size should be scaled more slowly than model size ($D \propto N^{0.74}$, $D$ is dataset size, $N$ is model size). Later, Hoffmann et al. (2022) contradicted this finding, showing that model size and dataset size should be scaled roughly equally for optimal scaling. They highlight the differences in setup which lead to them showing that large models should be

---

*Equal contribution

| SCALING LAW REPRODUCILIBILITY CHECKLIST | | | |
| --- | --- | --- | --- |
| **Scaling Law Hypothesis (§3)** | **Training Setup (§4)** | **Data Collection(§5)** | **Fitting Algorithm (§6)** |
| • Form | • # of models | • Checkpoint Open Sourcing | • Objective (Loss) |
| • Variables (Input) | • Model size range | • # Checkpoints per Power Law | • Algorithm |
| • Parameters | • Dataset source & size | | • Optimization Hyperparameters |
| • Derivation and Motivation | • Parameter/ FLOP Count Calculation | • Evaluation Dataset & Metric | • Optimization Initialization |
| • Assumptions | • Hyperparameter Choice | • Metric Modification & Code | • Data Usage Coverage |
| | • Other Settings | | • Validation of Law(s) |
| | • Code Open Sourcing | | |

Figure 1: We introduce a checklist for researcher to use for scaling laws research. In Appendix B, we include an expanded version of the checklist that may be used as a template.

trained for significantly more tokens: particularly, they point to using later checkpoints, training larger models, a different learning rate schedule and changing the number of training tokens used across runs. Multiple followup works have focused on either reproducing or explaining the differences between these two papers (Besiroglu et al., 2024; Porian et al., 2024; Pearce & Song, 2024b). The authors find it challenging to reproduce results of previous papers - we refer the reader to Section 2 for a further discussion on these replication efforts.

Motivated by this, we survey over 50 papers on scaling laws across a variety of modalities, tasks and architectures, and find that essential details needed to reproduce scaling law studies are often underreported. We broadly categorize these details as follows:

**Section 3: What *form* are we fitting?**  Researchers may choose any number of power law forms relating any set of variables, to which they fit the data extracted from training runs. Even seemingly minor differences in form, may imply critical changes in assumptions – for example, about certain interactions between variables which are excluded, the definitions of these variables or error terms which are deemed significant enough to include.

**Section 4: How do we *train models*?**  In order to fit a scaling law, one needs to train a range of models spanning orders of magnitude in parameter count and/or dataset size. Each model requires a multitude of hyperparameter and parameter choices, such as the specific model/dataset sizes to use, the architecture shape, batch size or learning rate schedule.

**Section 5: How do we *extract data* after training?**  Once these models are trained, downstream metrics like perplexity must be obtained from the intermediate or final checkpoints. This data may be also scaled, interpolated or bootstrapped to create more datapoints to fit the power law parameters.

**Section 6: How are we *optimizing* the fit?**  Finally, the variable must be fit with an objective and optimization method, which may in turn have their own initialization and hyperparameters to choose.

To aid scaling laws researchers in reporting details necessary to reproduce their work, we propose a checklist (Figure 1 - an expanded version may be found in Appendix B ). Based on this checklist, we summarize these details for all 51 papers in tabular form in Appendix C. We find that important details are frequently underreported, significantly impacting reproducibility, especially in cases where there is no code - only 19 of 42 papers surveyed have analysis code/code snippets available. Additionally, 23 (a little over half) of surveyed papers do not describe the optimization process, and 15 do not describe how training FLOPs or number of parameters are counted, which has been found to significantly change results (Porian et al., 2024). In addition, we fit our own power laws to further demonstrate how these choices critically impact the final scaling law (Section §7).

## 2 PAPERS ON SCALING LAWS

Researchers have proposed scaling laws to study the scaling of deep learning across multiple domains and for several tasks. Studies of the scaling properties of generalization error with training data size and model capacity predate modern deep learning. Banko & Brill (2001) observed a power law scaling of average validation error on a confusion set disambiguation task with increasing dataset size. The authors also claimed that the model size required to fit a given dataset grows log linearly. As

for larger scale models, Amodei et al. (2016) observe a power-law WER improvement on increasing training data for a 38M parameter Deep Speech 2 model. Hestness et al. (2017) show similar power law relationships across several domains such as machine translation, language modeling, image processing and speech recognition. Moreover, they find that these exponential relationships found hold across model improvements.

Kaplan et al. (2020) push the scale of these studies further, studying power laws for models up to 1.5B parameters trained on 23B tokens to determine the optimal allocation of a fixed compute budget. Later studies (Hoffmann et al., 2022; Hu et al., 2024) revisit this and find that Kaplan et al. (2020) greatly underestimate the amount of data needed to train models optimally, though major procedural differences render it challenging to attribute the source of this discrepancy. Since then, researchers have studied various aspects of scaling up language models. Wei et al. (2022) examine the emergence of abilities with scale that are not present in smaller models, while Hernandez et al. (2021) study the scaling laws for transfer between distributions in a finetuning setting. Henighan et al. (2020) consider possible interactions between different modalities while recently, Aghajanyan et al. (2023) study scaling in multimodal foundation models. Tay et al. (2022) show that not all architectures scale equally well, highlighting the importance of using scaling studies to guide architecture development. Poli et al. (2024) scale hybrid architectures like Mamba (Gu & Dao, 2023), showing the efficacy of this new model family. Other researchers also formulate specific scaling laws to study other Transformer based architectures. For example, Clark et al. (2022) and Frantar et al. (2023) introduce new scaling laws to study mixture of expert models (Fedus et al., 2022; Shazeer et al., 2017) and sparse models (Zhu & Gupta, 2017) respectively. Researchers have also used scaling laws to study encoder-decoder models for neural machine translation (Ghorbani et al., 2021; Gordon et al., 2021), and the effect of data quality and language on scaling coefficients (Bansal et al., 2022; Zhang et al., 2022). While language models form the majority of the papers surveyed, we also consider papers that study VLMs (Cherti et al., 2023; Henighan et al., 2020), vision (Alabdulmohsin et al., 2022; Zhai et al., 2022), reinforcement learning (Hilton et al., 2023; Jones, 2021; Gao et al., 2023) and recommendation systems (Ardalani et al., 2022). We further discuss different forms of scaling laws researchers introduce for the specific research questions they wish to answer in Section 3.

A majority of the surveyed papers study Transformer (Vaswani, 2017) based models, but a few consider different architectures. For example, Sorscher et al. (2022) investigate data pruning laws in ResNets, and some smaller scale studies use MLPs or SVMs (Hashimoto, 2021). This overrepresentation is perhaps partially a result of Transformer-based models achieving higher scale than other architectures; a ResNet101 has 44M parameters, while the largest Llama 3 model has 405B.

**Replication Efforts** Besiroglu et al. (2024) seek to reproduce the parameter fitting approach used by Hoffmann et al. (2022). They are unable to recover the scaling law from Hoffmann et al. (2022), and demonstrate that the claims of the original paper are inconsistent with descriptions of the setup. They then seek to improve the fit of the scaling law by initializing from the parameters found in Hoffmann et al. (2022) and modifying parts of the power law fitting process.

Porian et al. (2024) isolate several decisions as primarily responsible for the discrepancy between the recommendations of Kaplan et al. (2020) and Hoffmann et al. (2022): (1) learning rate scheduler warmup, (2) learning rate decay, (3) inclusion of certain parameters in total parameter count, and (4) specific training hyperparameters. By adjusting these factors, they are able to reproduce the results of Kaplan et al. (2020) and Hoffmann et al. (2022). However, they only use 16 training runs to fit their scaling laws, each designed to match one targeted setting (e.g., replicating Kaplan et al. (2020)). Instead of using raw loss values, they fit to loss values found by interpolating between checkpoints. Like Pearce & Song (2024b), they apply a log transform and linear regression to fit their law.

## 3 WHAT *form* ARE WE FITTING?

A majority of papers we study fit some kind of power law ($f(x) = ax^{-k}$). That is, they specify an equation defining the relationship between multiple factors, such that a proportional change in one results in the proportional change of at least one other. They then optimize this power law to find some parameters. A few efforts do not seem to fit a power law, but may show a line of best fit, obtained through unspecified methods (Rae et al., 2021; Dettmers et al., 2022; Tay et al., 2022; Shin et al., 2023; Schaeffer et al., 2023; Poli et al., 2024).

The specific form may be motivated by researcher intuition, previous empirical results, prior work, code implementation, or data availability. More importantly, the form is often determined by the specific question(s) a paper investigates. For example, one may attempt to predict the performance achieved by scaling up different model architectures, or the optimal ratio for model scaling vs data scaling when increasing training compute (Kaplan et al., 2020; Hoffmann et al., 2022). Based on this, we loosely classify scaling laws by their form as *performance prediction* and *ratio optimization* approaches. We indicate this classification for all surveyed papers in Appendix C.

## 3.1 RATIO OPTIMIZATION

The simplest scaling law forms usually predict the relation between two variables in an optimal setting. For example, approaches 1 and 2 from Hoffmann et al. (2022) fit to the optimal (i.e., lowest loss) $D$ and $N$ values for a particular compute budget $C$. Porian et al. (2024), aiming to resolve these inconsistencies, defines $\rho^* = \frac{D^*}{N^*}$ and writes this relationship as:

$$N^*(C) = N_0^* \cdot C^\alpha; D^*(C) = D_0^* \cdot C^\alpha; \rho^*(C) = \rho_0^* \cdot C^\alpha \tag{1}$$

They assume $C \approx 6ND$, and thus only need to fit the first equation; the other power laws can be inferred. This simplicity is deceptive in some cases, as collecting $(N^*(C), C)$ pairs may not be trivial. It is possible to fix $C$ and follow a binary search approach to train a multitude of models, then bisect to approximate the performance-optimal $N, D$ pair. However, this quickly grows prohibitively costly. In practice, it is common to interpolate between a set of fixed results to estimate the true $N^*(C)$ §5. This adds to the complexity of this approach, and introduces a hidden dependency on the performance evaluation, yet it does not actually predict the performance of the optimal points. If only the performance of the optimal-ratio model is of interest, it is possible to fit a second power law $L(N^*(C), D^*(C)) = a \cdot C^\alpha$. Most papers we survey choose to fit a power law which directly predicts performance.

## 3.2 PERFORMANCE PREDICTION

Kaplan et al. (2020) proposes a power law between Loss $L$, number of model Parameters $N$, and number of Dataset tokens $D$:

$$L(N, D) = \left[ \left( \frac{N}{N_c} \right)^{\frac{\alpha_N}{\alpha_D}} + \frac{D}{D_c} \right]^{\alpha_D} \tag{2}$$

On the other hand, Approach 3 of Hoffmann et al. (2022) proposes

$$L(N, D) = E + \frac{A}{N^\alpha} + \frac{B}{D^\beta} \tag{3}$$

In both of the above, all variables other than $L$, $N$, and $D$ are parameters to be found in the power law fitting process. Though these two forms are quite similar, they differ in some assumptions. Kaplan et al. (2020) constructs their form on the basis of 3 expected scaling law behaviors, and Hoffmann et al. (2022) explains in their Appendix D that their form is based on risk decomposition. The resulting Kaplan et al. (2020) form includes an interaction between $N$ and $D$ in order to satisfy a constraint requiring assymmetry introduced by one of their expected behaviors. The Hoffmann et al. (2022) form, on the other hand, consists of 3 additive sources of error, $E$ representing the irreducible error that would exist even with infinite data and compute budget, as well as two terms representing the error introduced by limited parameters and limited data, respectively.

Power laws for performance prediction can sometimes yield closed form solutions for optimal ratios as well. However, the additional parameters and input variables, introduced by the need to incorporate the performance metric term, add random noise and dimensionality. This increases the difficulty of optimization convergence, so when prediction performance is not the aim, a ratio optimization approach is frequently a better choice.

Many papers directly adopt one of these forms, but some adapt these forms to study relationships with other input variables. Clark et al. (2022), for example, study routed Mixture-of-Expert models, and propose a scaling law that relates dense model size (effective parameters) $N$ and number of experts $E$

with a biquadratic interaction ($\log L(N, E) \triangleq a \log N + b \log E + c \log N \log E + d$). Frantar et al. (2023) study sparsified models, and propose a scaling law with an additional parameter sparsity $S$, the optimal value of which increases with $N$ ($L(S, N, D) = \left(a_S(1-S)^{b_S} + c_S\right) \cdot \left(\frac{1}{N}\right)^{b_N} + \left(\frac{a_D}{D}\right)^{b_D} + c$). Other papers change the form to model variables in the data setup. Aghajanyan et al. (2023) consider interference and synergy between multiple data modalities ($L(N, D_j) = E_j + \frac{A_j}{N^{\alpha_j}} + \frac{B_j}{|D_j|^{\beta_j}}$, $L(N, D_i, D_j) = \left[\frac{L(N,D_i)+L(N,D_j)}{2}\right] - C_{i,j} + \frac{A_{i,j}}{N^{\alpha_{i,j}}} + \frac{B_{i,j}}{|D_i|+|D_j|^{\beta_{i,j}}}$), while Goyal et al. (2024), Fernandes et al. (2023) and Muennighoff et al. (2024) add terms to their scaling law formulations which represent mixing data sources and/or repeated data, using notions such as diminishing utility. A comprehensive list of the power law forms in the surveyed papers may be found in Table 6.

# 4 How do we *train models*?

In order to fit a scaling law, one needs to train a range of models across multiple orders of magnitude in model size and/or dataset size. Researchers must first decide the range and distribution of $N$ and $D$ values for their training runs, in order to achieve stable convergence to a solution with high confidence, while limiting the total compute budget of all experiments. Many papers did not specify the number of data points used to fit each scaling law; those that did range from 4 to several hundred, but most used fewer than 50 data points. The specific $N$ and $D$ values also skew the optimization process towards a certain range of $N/D$ ratios, which may be too narrow to include the true optimum. Some approaches, such as using IsoFLOPs (Hoffmann et al., 2022), additionally dictate rules for choosing $N$ and $D$ values. Moreover, using a minimum $N$ or $D$ value may result in outlier values that may need to be dropped (Porian et al., 2024; Shin et al., 2023; Henighan et al., 2020). We investigate this choice in Section §7.2

The definition of $N$, $D$, or compute cost $C$ can affect the results of a scaling study. For example, if a study studies variation in tokenizers, a definition of training data size based on character count may be more appropriate than one based on token count (Tao et al., 2024). The inclusion or exclusion of embedding layer compute and parameters, may also skew the results of a study - a major factor in the different in optimal ratios determined by Kaplan et al. (2020) and Hoffmann et al. (2022) has been attributed to not factoring embedding FLOPs into the final compute cost (Pearce & Song, 2024b; Porian et al., 2024). Given the increase in extremely long context models (128k-1M) Reid et al. (2024), the commonly used training FLOPs approximation $C = 6ND$ (see Appendix C) may not hold for such models, given the additional cost proportional to the context length and model dimension - Bi et al. (2024) introduce a new terms non-embedding FLOPs/token to account for this.

Scaling law fit depends on the performance of each individual checkpoint, which is highly dependent on factors such as training data source, architecture and hyperparameter choice. Bansal et al. (2022) and Goyal et al. (2024), for instance, discuss the effect of data quality and composition on power law exponents and constants. Repeating data has also been found to yield different scaling patterns in large language models (Muennighoff et al., 2024; Goyal et al., 2024).

Researchers have also studied the effect of architecture choice on scaling - Hestness et al. (2017) find that architectural improvements only shift the irreducible loss, while Poli et al. (2024) suggest that these improvements may be more significant. The way in which a model is scaled can also affect results. Within the same architecture family, Clark et al. (2022) show that increasing the number of experts in a routed language model has diminishing returns beyond a point, while Ghorbani et al. (2021) find that scaling the encoder and decoder have different effects on model performance. Scaling embedding size can also drastically change scaling trends (Tao et al., 2024).

The optimal hyperparameters to train a model changes with scale. Changing batch size, for example, can change model performance McCandlish et al. (2018); Kaplan et al. (2020). Optimal learning rate is another hyperparameter shown to change with scale, though techniques such as those proposed in Tensor Programs series of papers (Yang et al., 2022) can keep this factor constant with simple changes to initialization. More specifically, changing the learning rate schedule from a cosine decay to a constant learning rate with a cooldown (or even changing the learning rate hyperparameters) has been found to greatly affect the results of scaling laws studies (Hu et al., 2024; Porian et al., 2024; Hägele et al., 2024; Hoffmann et al., 2022).

One common motivation for fitting a scaling laws is extrapolation to higher compute budgets. However, there is no consensus on the orders of magnitude up that one can project a scaling law and still find it accurate, nor on the breadth of compute budgets that should be covered by the data. We find that the range of model size $N$ and dataset size $D$ greatly varies, with the maximum value of $N$ in each paper ranging from 10M parameters to around 7B and that of $D$ being as large as 400B tokens. For most papers we survey, the scales are relatively modest: 13 of 51 papers train models beyond 2B parameters; most only train models smaller than 1B parameters. It has been shown, with some controversy Schaeffer et al. (2023), that scaling to significantly larger scales can result in new abilities that did not appear in smaller models (Wei et al., 2022). Forecasting limits to extrapolation and the appearance of new abilities at new scales is an open question.

## 5  HOW DO WE *collect data* FROM MODEL TRAINING?

To evaluate the range of models trained to fit a scaling law, train or validation loss are most commonly used, but some works consider other metrics, such as ELO score (Jones, 2021; Neumann & Gros, 2022), reward model score (Gao et al., 2023), or downstream task metrics like accuracy or classification error rate (Henighan et al., 2020; Zhai et al., 2022; Cherti et al., 2023; Goyal et al., 2024; Gao et al., 2023). This choice is non-trivial - while some papers show that there is a power law relation between the predicted loss found by using validation loss and a different downstream task (Dubey et al., 2024), it is possible for the results of a study to change completely depending on the metric used. Schaeffer et al. (2023), for example, find that using linear metrics such as Token-edit distance instead of non-linear metrics such as accuracy produces smooth, continuous predictable changes in model performance, contrary to an earlier study by Wei et al. (2022). Moreover, Neumann & Gros (2022) find that they are unable to use test loss instead of Elo scores to fit a power law.

While it is most straightforward to evaluate only the final checkpoint on the target metric, some studies may use the median score of the last several checkpoints of each training run Ghorbani et al. (2021), or multiple intermediate checkpoints throughout each training run for various reasons. One common reason is that this is the only computationally feasible way to obtain a fit with sufficient confidence intervals (Besiroglu et al., 2024). For instance, the ISOFLop approach to finding the optimal $D/N$ ratio in Hoffmann et al. (2022) requires training multiple models for each targeted FLOP budget - this would be computationally prohibitive to do without using intermediate checkpoints. Hoffmann et al. (2022), in particular, use the last $15\%$ checkpoints. Some papers also report bootstrapping values (Ivgi et al., 2022). This detail is often not specified in scaling law papers, with only 29 of 51 papers reporting this information - we point the reader to Appendix C for an overview.

A related technique is performance interpolation. Porian et al. (2024) do not aim to exactly match the desired FLOP counts when evaluating model checkpoints mid-training. They instead interpolate between multiple model checkpoints to estimate the performance of a model with the target number of FLOPs. Hoffmann et al. (2022) and Tao et al. (2024) also interpolate intermediate checkpoints. Hilton et al. (2023), relatedly, smooth the learning curve before extracting metric scores.

As discussed in Section 4, training models with too little data or too few parameters can skew the results. To prevent this issue, several works report filtering out data points before fitting their power law. Henighan et al. (2020) drop their smallest models, while Hilton et al. (2023) and Hoffmann et al. (2022) exclude early checkpoints. Muennighoff et al. (2024) remove outlier datapoints that perform badly due to excess parameters or excess epochs. Similarly, Ivgi et al. (2022) remove outlier solutions after bootstrapping.

## 6  HOW ARE WE *optimizing* THE FIT?

The optimization of a power law requires several design decisions, including optimizer, loss, initialization values, and bootstrapping. We discuss each in this section. Over half of the papers we analyze do not provide any information about their power law fitting process, or provide limited information only and fail to detail crucial aspects. Specifically, many papers fail to describe their choice of optimizer or loss function. In Table 2, we provide an overview of the optimization details (if specified) for each paper considered.

| Paper | Curve-fitting Method | Loss Objective | Hyperparameters Reported? | Initialization | Are scaling laws validated? |
|---|---|---|---|---|---|
| Rosenfeld et al. (2019) | Least Squares Regression | Custom error term | N/A | Random | Y |
| Mikami et al. | Non-linear Least Squares in log-log space | | N/A | N/A | Y |
| Schaeffer et al. (2023) | NA | NA | NA | NA | NA |
| Sardana & Frankle (2023) | L-BFGS | Huber Loss | Y | Grid Search | N |
| Sorscher et al. (2022) | NA | NA | NA | NA | NA |
| Caballero et al. (2022) | Least Squares Regression | MSLE | N/A | Grid Search, optimize one | Y |
| Besiroglu et al. (2024) | L-BFGS | Huber Loss | Y | Grid Search | Y |
| Gordon et al. (2021) | Least Squares Regression | | N/A | N.S. | N |
| Bansal et al. (2022) | NS | NS | N | NS | N |
| Hestness et al. (2017) | NS | RMSE | N | NS | Y |
| Bi et al. (2024) | NS | NS | N | NS | Y |
| Bahri et al. (2024) | NS | NS | N | NS | N |
| Geiping et al. (2022) | Non-linear Least Squares | | NA | Non-augmented parameters | Y |
| Poli et al. (2024) | NS | NS | N | NS | N |
| Hu et al. (2024) | scipy curvefit | NS | N | NS | N |
| Hashimoto (2021) | Adagrad | Custom Loss | Y | Xavier | Y |
| Ruan et al. (2024) | Linear Least Squares | Various | N/A | N/A | Y |
| Anil et al. (2023) | Polynomial Regression (Quadratic) | N.S. | N | N.S. | Y |
| Pearce & Song (2024a) | Polynomial Least Squares | MSE on Log-loss | N/A | N/A | N |
| Cherti et al. (2023) | Linear Least Squares | MSE | N/A | N/A | N |
| Porian et al. (2024) | Weighted Linear Regression | weighted SE on Log-loss | N/A | N/A | Y |
| Alabdulmohsin et al. (2022) | Least Squares Regression | MSE | Y | N.S. | Y |
| Gao et al. (2024) | N.S | N.S | N.S | N.S | N.S |
| Muennighoff et al. (2024) | L-BFGS | Huber on Log-loss | Y | Grid Search, optimize all | Y |
| Rae et al. (2021) | None | None | N/A | N/A | N |
| Shin et al. (2023) | NA | NA | NA | NA | NA |
| Hernandez et al. (2022) | NS | NS | NS | NS | NS |
| Filipovich et al. (2022) | NS | NS | NS | NS | NS |
| Neumann & Gros (2022) | NS | NS | NS | NS | NS |
| Droppo & Elibol (2021) | NS | NS | NS | NS | NS |
| Henighan et al. (2020) | NS | NS | NS | NS | NS |
| Goyal et al. (2024) | Grid Search | L2 error | Y | NA | Y |
| Aghajanyan et al. (2023) | L-BFGS | Huber on Log-loss | Y | Grid Search, optimize all | Y |
| Kaplan et al. (2020) | NS | NS | NS | NS | N |
| Ghorbani et al. (2021) | Trust Region Reflective algorithm, Least Squares | Soft-L1 Loss | Y | Fixed | Y |
| Gao et al. (2023) | NS | NS | NS | NS | Y |
| Hilton et al. (2023) | CMA-ES+Linear Regression | L2 log loss | Y | Fixed | Y |
| Frantar et al. (2023) | BFGS | Huber on Log-loss | Y | N Random Trials | Y |
| Prato et al. (2021) | NS | NS | NS | NS | NS |
| Covert et al. (2024) | Adam | Custom Loss | Y | NS | Y |
| Hernandez et al. (2021) | NS | NS | NS | NS | Y |
| Ivgi et al. (2022) | Linear Least Squares in Log-Log space | MSE | NA | NS | Y |
| Tay et al. (2022) | NA | NA | NA | NA | NA |
| Tao et al. (2024) | L-BFGS, Least Squares | Huber on Log-loss | Y | N Random Trials from Grid | Y |
| Jones (2021) | L-BFGS | NS | NS | NS | NS |
| Zhai et al. (2022) | NS | NS | NS | NS | NS |
| Dettmers & Zettlemoyer (2023) | NA | NA | NA | NA | NA |
| Dubey et al. (2024) | NS | NS | NS | NS | Y |
| Hoffmann et al. (2022) | L-BFGS | Huber on Log-loss | Y | Grid Search, optimize all | Y |
| Ardalani et al. (2022) | NS | NS | NS | NS | NS |
| Clark et al. (2022) | L-BFGS-B | L2 Loss | Y | Fixed | NS |

Table 2: We provide an overview of which papers provide specific details required to reproduce how they fit their scaling law equation.

**Optimizer** Power laws are most commonly fit with a variety of algorithms designed to optimize non-linear functions. One of the most common is the BFGS (Broyden-Fletcher-Goldfarb-Shanno) algorithm, or a variation L-BFGS (Liu & Nocedal, 1989). Some papers (Hashimoto, 2021; Covert et al., 2024) use Adam, Adagrad, or other optimizers common in machine learning, such as AdamW, RMSProp, and SGD. Though effective for LLM training, these are sometimes ill-suited for the purpose of fitting a scaling law, due to various factors limiting their practicality, such as data-hungriness. Goyal et al. (2024) forgo the use of an optimizer altogether due to instability of solutions (see initialization) and rely exclusively on grid search to fit their scaling law parameters.

Some scaling law works (Rosenfeld et al., 2019) opt to use a linear method, such as linear regression, which is generally much simpler. To do this, they typically convert the hypothesized power law to a linear form by taking the log. For example, for a power law $y^b = c \cdot x^a + d$, use the form $\beta \cdot log(y) = \gamma + \alpha \cdot log(x)$ instead. For loss prediction, this results in a form similar to $log(L(N, D)) = \alpha \cdot logN + \beta \cdot logD + E$. This trick is employed even when using an optimizer capable of operating on non-linear functions (Hashimoto, 2021). Though this conversion may sometimes work in practice, it is generally not advised because the log transformation also changes the distribution of errors, exaggerating the effects of errors at small values. This mismatch increases the likelihood of a poor fit (Goldstein et al., 2004). We found this approach to be very common among the papers we study.

**Loss** Various loss functions have been chosen for power law optimization, including variants on MAE (mean absolute error), MSE (mean squared error), and the Huber loss (Huber, 1992), which is identical to MSE for errors less than some value $\delta$ (a hyperparameter), but grows linearly, like MAE, for larger errors, effectively balancing the weighting of small errors with robustness to outliers. Of the papers which specify their loss function, most use a variant of MAE (Ghorbani et al., 2021), MSE (Goyal et al., 2024; Hilton et al., 2023), Huber loss (Hoffmann et al., 2022; Aghajanyan et al., 2023; Frantar et al., 2023; Tao et al., 2024; Muennighoff et al., 2024), or a custom loss (Covert et al., 2024).

**Initialization** Initialization can have a substantial impact on final optimization fit (§7). One approach is to iteratively train with different initializations, selecting the best fit at the termination of the search. This is typically a grid search over choices for each parameter (Aghajanyan et al., 2023; Muennighoff et al., 2024), or a random sample from that grid (Frantar et al., 2023; Tao et al., 2024). Alternatively, the full grid of potential initializations can be evaluated on the loss function without training, and the most optimal $k$ used for initialization and optimization (Caballero et al., 2022). Finally, if a hypothesis exists, either from prior work or expert knowledge about the function, this hypothesis may be used instead of a search, or to guide the search (Besiroglu et al., 2024).

**Validating the Scaling Law** A majority of the papers surveyed do not report validating the scaling law in any meaningful way. Knowing this is critical to understanding whether the results of the scaling laws study are valid, given the examples given throughout the paper of scaling laws study conclusions changing depending on the process details. Porian et al. (2024) and Alabdulmohsin et al. (2022) use confidence intervals and goodness of fit measures to validate their scaling laws. Ghorbani et al. (2021) and Bansal et al. (2022) also do this. Otherwise, a majority of the papers that we report as validating their scaling laws mainly extrapolate to models a few orders of magnitude larger and observe the adherence to the scaling law obtained.

# 7 OUR REPLICATIONS AND ANALYSES

Each of the choices discussed above in Sections 3 - 6 may have a crucial impact on the result, yet it remains common to critically underspecify the setup for fitting a power law. Scaling law works often fail to open source their model and code, making reproduction infeasible, and likely contributing to contradictory conclusions as discussed in Section 2. Though some efforts have been made (Porian et al., 2024; Besiroglu et al., 2024) to reconcile such discrepancies, there is still only sparse understanding of the impact of each of the decisions we discuss.

To investigate the significance of these scaling law optimization decisions, we vary these choices to fit our own scaling laws. We fit both the Chinchilla-scraped data from Besiroglu et al. (2024), and data from our own models.

**Reconstructed Chinchilla data (Besiroglu et al., 2024)** This data is extracted from a vector-based figure in the pdf of Hoffmann et al. (2022), who claim that this includes all models trained for the paper. It consists of 245 datapoints, each corresponding to a final checkpoint collected at the end of model training. There is a potential risk of errors in this recovery process.

**Data from Porian et al. (2024)** This data includes training losses for Transformer LMs ranging in size from 5 to 901 million parameters, each trained on OpenWebText2 (Gao et al., 2020) and RefinedWeb data (Penedo et al., 2023), across a variety of data and compute budgets, from 5 million to 14 billion tokens, depending on parameter count. Each model is trained with a different peak learning rate and batch size setting, found by fitting a separate set of scaling laws.

**Our models** We train a variety of Transformer LMs, ranging in size from 12 million to 1 billion parameters, on varied data and compute budgets and hyperparameter settings. Details about our setup, including hyperparameters, are listed in Appendix A. We open source all of our models, evaluation results, code, and FLOP calculator at `https://github.com/hadasah/scaling_laws`

We fit a multitude of power laws and study the effects of: (1) power law form; (2) model learning rate; (3) compute budget, model size, and data budget range and coverage; (4) definition of $N$ and $C$; (5) inclusion of mid-training checkpoints; (6) power law parameter initialization; (7) choice of loss and (8) optimizer.

We plot all results in Appendix Figure 2. In our initial analysis of (5), we find substantial differences, sometimes improvements, in stability of fit when including mid-training checkpoints. As a result, we include analyses and corresponding figures for power law fitting with only data from final checkpoints, as well as with all mid-training checkpoints, when applicable.

Based on our observations, we also make some more concrete recommendations in Appendix §D, with the caveat that following the recommendations cannot guarantee a good scaling law fit.

## 7.1 FORM (§3)

We consider the (1) baseline Hoffmann et al. (2022) form (Approach 3) and then (2) apply the trick employed by Muennighoff et al. (2024) of setting $\alpha = \beta$ in the Hoffmann et al. (2022) form, which assumes the optimal $D/N$ ratio stays roughly constant $- L(N, D) = E + \frac{A}{N^\alpha} + \frac{B}{D^\alpha}$. We also compare with (3) an ISOFlop approach (Approach 3 of Hoffmann et al. (2022)), in which we fit an optimal $N$ for each compute budget $C$, $N_{opt}(C)$, which can then be used to fit the predicted loss $L(N, D)$. This approach usually necessitates the usage of mid-training checkpoints (discussed further in §7.3), as it is infeasible to train a large enough number of models for each FLOP budget considered. However, we apply it here without using only final model checkpoints, and extend to mid-training checkpoints in §7.3). We adapt the implementation from Porian et al. (2024), which contains more details about interpolation of data points and specific hyperparameters. In all data sets, (2) approaches the law reported by Hoffmann et al. (2022), but (1) only does so for the data from Porian et al. (2024) (Figure 2a).

We experiment with the power law form reported by Kaplan et al. (2020), but this consistently yields a law which suggests that the optimal number of data points is 1, even when varying many aspects of the power law fitting procedure. The difficulty of fitting to this form might be partially a result of severe under-reporting in Kaplan et al. (2020) with regard to procedural details, including hyperparameters for both model training and fitting.

## 7.2 TRAINING (§4)

Hu et al. (2024) study the effects of several model training decisions, including batch size, learning rate schedule, and model width. Their analysis focuses on optimizing hyperparameters, not on the ways hyperparameter and architecture choices affect the reliability of scaling law fitting. Observed variations between settings suggest that suboptimal performance could skew the scaling law fit.

To substantiate this further, we simulate the effects of not sweeping the learning rate in our models. As a baseline, (1) we sweep at each $(N, D)$ pair for the optimal learning rate over a range of values, at most a multiple of 2 apart. Next, (2) we use a learning rate of 1e-3 for all $N$, the optimal for our 1 billion parameter models, and do the same for (3) 2e-3 and (4) 4e-3, which is optimal for our 12 million parameter models. Lastly, we use all models across all learning rates at the same $N$ and $D$. Results vary dramatically across these settings. Somewhat surprisingly, using all learning rates results in a very similar power law to sweeping the learning rate, whereas using a fixed learning rate of 1e-3 or 4e-3 yields the lowest optimization loss or closest match to the Hoffmann et al. (2022) power laws, respectively (Figure 2b).

We also study the effects of limiting model scale range, data scale range (implicitly), and data-to-parameters range by filtering all datasets: we compare (1) using all $N, D$ scales, (2) only models with $N$ up to about 100 million or (3) 400 million parameters, and including (4) only models with $D/N \leq 18$ or (5) $D/N \geq 22$. These ranges are designed to exclude $D/N = 20$, the rule of thumb based on Hoffmann et al. (2022). The minimum or maximum $D/N$ ratio tested does skew results; above $10^{22}$ FLOPs, (4) and (5) fit to optimal ratios $D/N < 18$ and $D/N > 22$, respectively. Removing our largest models in (2) also creates a major shift in the predicted optimal $D/N$ (Figure 2c).

Related to choice of $N, D$ and $C$ values, we investigate different ways of counting $N$ and $C$, specifically whether to include embedding parameters and FLOPs. We compare (1) our baseline, which includes embeddings in both $N$ and $C$, with (2) excluding embeddings only in $N$, (3) excluding embeddings only in $C$, (4) excluding embeddings in both $N$ and $C$. We also compare to (5) using the $C = 6ND$ approximation, including embedding parameters. In all other settings, we calculate the FLOPs in a manner similar to Hoffmann et al. (2022), and we open source the code for these calculations. With both datasets, the exclusion of embeddings in FLOPs has very little impact on the final fit. Similarly, using the $C = 6ND$ approximation has no visible impact. For the Porian et al. (2024) models, the exclusion of embedding parameters in the calculation of $N$ results in scaling laws which differ substantially, and with increasing divergences at large scales (Figure 2d).

### 7.3 DATA COLLECTION (§5)

We compare using data from (1) only the final checkpoints of each model with (2) using all available mid-training checkpoints. We also consider removing checkpoints collected during the (3) first 10%, (4) 20%, and (5) 50% of training, Using mid-training checkpoints sometimes results in more stable fits which are similar to the Hoffmann et al. (2022) scaling laws, but the effect is unreliable and is dependent on other decisions. Following this finding, we also run all other analyses using setting (2), and find that these scaling law fits are often more consistent when varying other aspects of the fitting procedure (Figure 2e).

### 7.4 FITTING (§6)

We vary the initialization method for our power law fitting procedure: (1) our baseline replication of Hoffmann et al. (2022) Approach 3, which conducts the full optimization process on a grid of 6x6x5x5x5=4500 initializations (Hoffmann et al., 2022), (2) searching for the lowest loss initialization point (Caballero et al., 2022) and optimizing only from that initialization, (3) optimizing from the 1000 lowest loss initializations, (4) randomly sampling $k(=100)$ points (Frantar et al., 2023; Tao et al., 2024), and (5) initializing with the coefficients found in Hoffmann et al. (2022), as Besiroglu et al. (2024) does. With the Besiroglu et al. (2024) data, (5) yields a fit nearly identical to that reported by Hoffmann et al. (2022), although (1) results in the lowest fitting loss. With the Porian et al. (2024) data, all approaches except (2) yield a very similar fit, which gives a recommended $D/N$ ratio similar to that of Hoffmann et al. (2022). However, using our data, (2) optimizing over only the most optimal initialization yields the best match to the Hoffmann et al. (2022) power laws, followed by (5) initialization from the reported Hoffmann et al. (2022) scaling law parameters. Optimizing over the full grid yields the power law that diverges most from the Hoffmann et al. (2022) law, suggesting the difficulty of optimizing over this space, and the presence of many local minima (Figure 2f).

Then, we analyze the choice of loss objectives, including (1) the baseline log-Huber loss, (2) MSE, (3) MAE, and (4) the Huber loss. We find that there is somewhat less variance in the resulting power laws than we have seen in other power law fitting decisions, but the recommended optimal data budgets still span a wide range (Figure 2g).

Finally, we consider the choice of optimizer. We first fit a power law to using the original (1) L-BFGS, with settings matching those described by Hoffmann et al. (2022). L-BFGS and BFGS implementations have an early stopping mechanism, which conditions on the stability of the solution between optimization steps. We experiment with setting this threshold for L-BFGS to a (2) higher value 1e-4 (stopping earlier). We find that using any higher or lower values results in the same solutions or instability for all three datasets. L-BFGS and BFGS also have the option to use the true gradient of the loss, instead of an estimate, which is the default. In this figure, we include this setting, (3) using the true gradient for L-BFGS. We compare these L-BFGS settings to (4) BFGS. We test the same tolerance value and gradient settings, and find that none of these options change the outcome of BFGS in our analysis. Finally, we compare to (5) non-linear least squares and (6) pure grid search, using a grid 5 times more dense along each axis as that which we used for initialization with other optimizers. This density is chosen to approximately match the runtime of L-BFGS. Many of these optimizer settings converge to similar solutions, but this depends on the data, and the settings which diverge from this majority do not follow any immediately evident pattern (Figure 2h).

## 8 CONCLUSION

We survey over 50 papers on scaling laws, and discuss differences in form, training setup, evaluation, and curve fitting, which may lead to significantly different conclusions. We also discuss significant under-reporting of details crucial to replicating the conclusion of these studies, and provide guidelines in the form of a checklist aid researchers in reporting more complete details. In addition to discussing several prior replication studies in literature, we empirically demonstrate the fragility of this process, by systematically varying these choices on available checkpoints and models that we train from scratch. We choose to avoid overly prescriptive recommendations, because there is no known set of actions which can guarantee a good scaling law fit, but we make some suggestions based on patterns in our findings (Appendix §D). Despite our preliminary investigations, our understanding of which decisions may skew the results of a scaling law study is sparse, and defines the path for future work.

**Ethics Statement**  This work discusses how a lack of reproducibility and open-sourcing may be harmful for scaling laws research, given that the factors in a study setup that may change research conclusions vary widely.

**Reproducibility Statement**  The model checkpoints and analysis code required to reproduce the results discussed in Section 7 are at `https://github.com/hadasah/scaling_laws`.

ACKNOWLEDGMENTS

We thank Aaron Defazio, Divyansh Pareek, Aditya Kusupati and Tim Althoff for their valuable feedback. We also acknowledge the computing resources and support from the Hyak supercomputer system at the University of Washington.

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

# A    OUR MODEL TRAINING (§7)

We train a variety of Transformer LMs, ranging in size from 12 million to 1 billion parameters, on FineWeb (Penedo et al., 2024), tokenized with the GPT-NeoX-20B tokenizer (Black et al., 2022), which is a BPE tokenizer with a vocabulary size of 50257, trained on the Pile (Gao et al., 2020). All models were trained on a combination of NVIDIA GeForce RTX 2080 Ti, Quadro RTX 6000, NVIDIA A40, and NVIDIA L40 GPUs. These transformers follow the standard architecture, and use pre-layer RMSNorm (Kudo, 2018), the SwiGLU activation function (Shazeer, 2020), and rotary positional embeddings (Su et al., 2024). We use a batch size of 512 with a sequence length of 2048. The learning rate is warmed up linearly over 50 steps to the peak learning rate and then follows a cosine decay to 10% of the peak. We use an Adam optimizer with $\beta_1 = 0.9, \beta_2 = 0.95$. We sweep over learning rates and data budgets at each model scale. For our evaluation metric, we use perplexity on a validation set of C4 (Raffel et al., 2020). See Table 3 and Table 10 for additional architecture details and hyperparameters, including data budget and learning rate.

| Hyperparameter | Value |
|---|---|
| Vocabulary size | 50K |
| Batch Size | 512 |
| Sequence Length | 2048 |
| Attention Head Size | 64 |
| Learning Rate | Swept $2^{\{0,1,2,3\}} \cdot 10^{\{-3,-4\}}$ |
| Feedforward Dimension | 4 x hidden dimension |
| LR Schedule | Cosine Decay |
| LR Warmup | 50 steps |
| End LR | 0.1 x Peak LR |
| Optimizer | Adam ($\beta_1, \beta_2 = 0.9, 0.95$) |

Table 3: Hyperparameter details for models trained in §7

| Model size | # layers | Hidden Dim | # Attn Heads | # Steps |
|---|---|---|---|---|
| 12M | 5 | 448 | 7 | {100, 200, 250, 360, 500, 750, 1,000, 4,000} |
| 17M | 7 | 448 | 7 | {100, 200, 250, 500, 750, 1,000, 1,250, 1,500} |
| 25M | 8 | 512 | 8 | {250, 360, 500, 750, 1,000, 1,500, 2,000, 8,000, 16,000} |
| 35M | 9 | 576 | 9 | {200, 250, 360, 500, 750, 1,000, 1,250, 1,500, 4,000, 16,000} |
| 50M | 10 | 640 | 10 | {250, 500, 750, 1,000, 1,250, 1,500, 1,800, 2,000, 2,500, 4,000, 16,000} |
| 70M | 12 | 704 | 11 | {250, 500, 750, 1,000, 1,250, 1,500, 2,000, 2,500, 3,000, 5,000} |
| 100M | 14 | 768 | 12 | {250, 500, 1,000, 1,500, 2,000, 2,500, 3,000, 4,000, 6,000, 12,000} |
| 200M | 18 | 960 | 15 | {500, 1,500, 6,000, 9,000} |
| 300M | 19 | 1152 | 18 | {4,000} |
| 400M | 20 | 1280 | 20 | {3,000, 6,000} |
| 1B | 26 | 1792 | 28 | {20,000} |

Table 4: Architecture and Data budget details for models trained in §7

## B    FULL CHECKLIST

We define each category as follows:

- **Scaling Law Hypothesis:** This specifies the form of the scaling law, that of the variables and parameters, and the relation between each.

- **Training Setup:** This specifies the exact training setup of each of the models trained to test the scaling law hypothesis.

- **Data Collection:** Evaluating various checkpoints of our trained models to collect data points that will be used to fit a scaling law in the next stage.

- **Fitting Algorithm:** Using the data points collected in the previous stage to optimize the scaling law hypothesis.

---

SCALING LAW REPRODUCILIBILITY CHECKLIST

**Scaling Law Hypothesis (§3)**
- What is the form of the power law?

- What are the variables related by (included in) the power law?

- What are the parameters to fit?

- On what principles is this form derived?

- Does this form make assumptions about how the variables are related?

**Training Setup (§4)**
- How many models are trained?

- At which sizes?

- On how much data each? On what data? Is any data repeated within the training for a model?

- How are model size, dataset size, and compute budget size counted? For example, how are parameters of the model counted? Are any parameters excluded (e.g., embedding layers)? How is compute cost calculated?

- Are code/code snippets provided for calculating these variables if applicable?

- How are hyperparameters chosen (e.g., optimizer, learning rate schedule, batch size)? Do they change with scale?

- What other settings must be decided (e.g., model width vs. depth)? Do they change with scale?

- Is the training code open source?

**Data Collection(§5)**
- Are the model checkpoints provided openly?

- How many checkpoints per model are evaluated to fit each scaling law? Which ones, if so?

- What evaluation metric is used? On what dataset?

- Are the raw evaluation metrics modified? Some examples include loss interpolation, centering around a mean or scaling logarithmically.

- If the above is done, is code for modifying the metric provided?

**Fitting Algorithm (§6)**
- What objective (loss) is used?

- What algorithm is used to fit the equation?

- What hyperparameters are used for this algorithm?

- How is this algorithm initialized?

- Are all datapoints collected used to fit the equations? For example, are any outliers dropped? Are portions of the datapoints used to fit different equations?

- How is the correctness of the scaling law considered? Extrapolation, Confidence Intervals, Goodness of Fit?

---

## C  FULL SHEET

We provide an overview of all the papers surveyed in Tables 5,6, 7, 8 and 9.

| Paper | Domain | Training Code? | Analysis Code? | Checkpoints? | Metric Scores? |
|---|---|---|---|---|---|
| Rosenfeld et al. (2019) | Vision, LM | N | N | N | N |
| Mikami et al. | Vision | N | Y | Y | Y |
| Schaeffer et al. (2023) | LM | N | N | N | N |
| Sardana & Frankle (2023) | LM | N | N | N | N |
| Sorscher et al. (2022) | Vision | N | N | N | Y |
| Caballero et al. (2022) | LM | N | Y | N | Y |
| Besiroglu et al. (2024) | LM | | Y | N | Y |
| Gordon et al. (2021) | NMT | Y | Y | Y | Y |
| Bansal et al. (2022) | NMT | N | N | N | N |
| Hestness et al. (2017) | NMT, LM, Vision, Speech | N | N | N | N |
| Bi et al. (2024) | LM | N | N | N | N |
| Bahri et al. (2024) | Vision | N | N | N | N |
| Geiping et al. (2022) | Vision | Y | Y | N | N |
| Poli et al. (2024) | LM | N | N | N | N |
| Hu et al. (2024) | LM | Y | N | N | N |
| Hashimoto (2021) | NLP | N | N | N | N |
| Ruan et al. (2024) | LM | Y | Y | N | Y |
| Anil et al. (2023) | LM | N | N | N | N |
| Pearce & Song (2024a) | LM | N | Y | N | N |
| Cherti et al. (2023) | VLM | Y | Y | Y | Y |
| Porian et al. (2024) | LM | Y | Y | N | Y |
| Alabdulmohsin et al. (2022) | LM, Vision | N | Y | Y | Y |
| Gao et al. (2024) | NLP | Y | Y | Y | N |
| Muennighoff et al. (2024) | LM | Y | Y | Y | N |
| Rae et al. (2021) | LM | N | N | N | N |
| Shin et al. (2023) | RecSys | N | N | N | N |
| Hernandez et al. (2022) | LM | N | N | N | N |
| Filipovich et al. (2022) | LM | N | N | N | N |
| Neumann & Gros (2022) | RL | Y | Y* | Y | N |
| Droppo & Elibol (2021) | Speech | N | N | N | N |
| Henighan et al. (2020) | LM, Vision, Video, VLM | N | N | N | N |
| Goyal et al. (2024) | LM, Vision, VLM | N | Y | N | Y |
| Aghajanyan et al. (2023) | Multimodal LM | N | N | N | N |
| Kaplan et al. (2020) | LM | N | N | N | N |
| Ghorbani et al. (2021) | NMT | N | Y | N | N |
| Gao et al. (2023) | RL/LM | N | N | N | N |
| Hilton et al. (2023) | RL | N | N | N | N |
| Frantar et al. (2023) | LM, Vision | N | N | N | N |
| Prato et al. (2021) | Vision | Y* | Y | Y | Y |
| Covert et al. (2024) | LM | Y | Y | N | N |
| Hernandez et al. (2021) | LM | N | N | N | N |
| Ivgi et al. (2022) | NLP | N | N | N | N |
| Tay et al. (2022) | LM | N | N | N | N |
| Tao et al. (2024) | LM | N | Y | N | Y |
| Jones (2021) | RL | Y | Y | N | Y |
| Zhai et al. (2022) | Vision | Y | N | N | N |
| Dettmers & Zettlemoyer (2023) | LM | N | N | N | N |
| Dubey et al. (2024) | LM | N | N | N | N |
| Hoffmann et al. (2022) | LM | N | N | N | N |
| Ardalani et al. (2022) | RecSys | N | N | N | N |
| Clark et al. (2022) | LM | N | Y | N | Y |

Table 5: Details on domain of experiments and availability of code by category for each paper surveyed.

| Paper | Power Law Form | Purpose Of Power Law (E.G., Performance Prediction, Optimal Ratio) | # Power Law Parameters | # Of Scaling Laws |
|---|---|---|---|---|
| Rosenfeld et al. (2019) | $\hat{\varepsilon}(m,n) = an^{-\alpha} + bm^{-\beta} + c_\infty$ | Performance Prediction | 5-6 | 8 |
| Mikami et al. | $L(n,s) = \delta(\gamma + n^{-\alpha})s^{-\beta}$ | Performance Prediction | 4 | 3 |
| Schaeffer et al. (2023) | None | N/A | NA | NA |
| Sardana & Frankle (2023) | $L(N,D) = E + \frac{A}{N^\alpha} + \frac{B}{D^\beta}; N^*(\ell, D_{inf}), D_\ell^*(\ell, D_{inf}) = \arg\min_{N,D_\ell\mid L(N,D_{inf})=\ell}$ | Performance Prediction | 5 | 4 |
| Sorscher et al. (2022) | $c\cdot\alpha^{-\beta}, c\cdot\exp(-b\alpha)$ | Performance Prediction | 2 | 34 |
| Caballero et al. (2022) | $y = a + (bx^{-c_0})\prod_{i=1}^{n}\left(1+\left(\frac{x}{d_i}\right)^{1/f_i}\right)^{-c_i * f_i}$ | Performance Prediction | 5+ | 100+ |
| Besiroglu et al. (2024) | $L(N,D) = E + \frac{A}{N^\alpha} + \frac{B}{D^\beta}$ | Performance Prediction | 5 | 1 |
| Gordon et al. (2021) | $L(D) = \left[\left(\frac{N}{N_c}\right)^{\frac{\alpha_N}{\alpha_D}} + \frac{D_c}{D}\right]^{\alpha_D}$ | Performance Prediction | 4 | 3 |
| Bansal et al. (2022) | $L(D) = \alpha(\tilde{D}^{-1} + C)^p$ | Performance Prediction | 3 | 20 |
| Hestness et al. (2017) | $\varepsilon(m) \sim \alpha m^{\beta_g} + \gamma$ | Performance Prediction | 3 | 17 |
| Bi et al. (2024) | $M_{opt} = M_{base}\cdot C^a, \eta_{opt} = 0.3118\cdot C^{-0.1250}$ $D_{opt} = D_{base}\cdot C^b, B_{opt} = 0.2920\cdot C^{0.3271}$ | Optimal Ratio, Performance Prediction | 2 | 5 |
| Bahri et al. (2024) | $L(D)\propto D^{-\alpha_D}, L(P)\propto P^{-\alpha_K}$ | Performance Prediction | 2 | 35 |
| Geiping et al. (2022) | $f(x) = ax^{-c} + b, v_{\text{Effective Samples from Augmentations}}(x) = f_{ref}^{-1}(f_{aug}(x)) - x$ | Performance Prediction | 3 | 50 |
| Poli et al. (2024) | $\log N^* \propto a\log C$ and $\log D^* \propto b\log C$ | Performance Prediction | 2 | |
| Hu et al. (2024) | $L(N,D) = C_N N^{-\alpha} + C_D D^{-\beta} + L_0$ | Performance Prediction | 5 | 6 |
| Hashimoto (2021) | $\min_{\lambda,\alpha}\mathbb{E}_{\hat{q},\hat{n}}\left[\left(\log(R(\hat{n},\hat{q}) - \epsilon) - \alpha\log(\hat{n}) + \log(C_\lambda(\hat{q}))\right)^2\right] R(\hat{n},\hat{q}) = \mathbb{E}\left[\ell\left(\hat{\theta}(p_{\hat{n},\hat{q}}); x,y\right)\right]$ | Performance Prediction | 2+n(data mixes) | 4 |
| Ruan et al. (2024) | $E_m \approx h\bar{\sigma}(\bar{\beta}^\top S_m + \epsilon)$ | Performance Prediction | 3 | |
| Anil et al. (2023) | $N^*_{ce} \approx N_0^*\cdot C^\alpha$ | Performance Prediction | 2 | 1 |
| Pearce & Song (2024a) | $N^*_{ce} = bC^m, L = bC^m$ | Optimal Ratio, Performance Prediction | 2 | 1 |
| Cherti et al. (2023) | $E = \beta C^\alpha$ | Performance Prediction | 2 | 8 |
| Porian et al. (2024) | $N^*(C) \approx N_0^*\cdot C^\alpha$ | Optimal Ratio | 2 | 6 |
| Alabdulmohsin et al. (2022) | $\varepsilon_x - \varepsilon_\infty = \beta x^c; \varepsilon_x - \varepsilon_\infty = \beta x^c; \varepsilon_x = \beta(x^{-1}+\gamma)^{-c}; \varepsilon_x = \gamma(x)(1+\gamma(x))^{-1}\varepsilon_0 + (1+\gamma(x))^{-1}\varepsilon_\infty$ | Performance Prediction | 2-4 | 600 |
| Gao et al. (2024) | $L(n,k) = \exp(\alpha + \beta_k\log(k) + \beta_n\log(n) + \gamma\log(k)\log(n)) + \exp(\zeta + \eta\log(k))$ | Performance Prediction | 2-6 | 1 |
| Muennighoff et al. (2024) | $L(U_N, U_D, R_N, R_D) = \frac{A}{\left(U_N + U_N R_N^*\left(1 - e^{\frac{-R_N}{R_N^*}}\right)\right)^\alpha} + \frac{B}{\left(U_D + U_D R_D^*\left(1 - e^{\frac{-R_D}{R_D^*}}\right)\right)^\beta} + E$ | Performance Prediction | 2 (+4) | 1 |
| Rae et al. (2021) | None | Performance Prediction | N/A | N/A |
| Shin et al. (2023) | None | Scaling trend | NA | NA |
| Hernandez et al. (2022) | $E = k + N^\alpha$ | Optimal Ratio | 2 | 1 |
| Filipovich et al. (2022) | $\mathcal{L}(C) = (C_c C)^{\alpha_C}$ | Performance Prediction | 2 | 3 |
| Neumann & Gros (2022) | $N_{opt}(C) = \left(\frac{C}{C_0}\right)^{\alpha_C^{pt}}, E_i = \frac{1}{1+(N_f/N_i)^{pN}}$ | Performance Prediction | 2 | 3 * 2 |
| Droppo & Elibol (2021) | $L(N,D) = \left[(L_\infty)^{\frac{1}{\alpha}} + \left(\frac{N_c}{N}\right)^{\frac{\alpha_N}{\alpha}} + \left(\frac{D_c}{D}\right)^{\frac{\alpha_D}{\alpha}}\right]^\alpha$ | Performance Prediction | 6 | 3 |
| Henighan et al. (2020) | $L(x) = L_\infty + \left(\frac{x_0}{x}\right)^{\alpha_x}$ | Performance Prediction | 3 | 36 |
| Goyal et al. (2024) | $y_k = a\cdot n_1^{b_1}\prod_{j=2}^{k}\left(\frac{n_j}{n_{j-1}}\right)^{b_j} + d$ | Performance Prediction | 2+ 2*n(data mixes) | 1 |
| Aghajanyan et al. (2023) | $L(N,D_j) = E_j + \frac{A_j}{N^{\alpha_j}} + \frac{B_j}{|D_j|^{\beta_j}}, L(N,D_i,D_j) = \left[\frac{L(N,D_i)+L(N,D_j)}{2}\right] - C_{i,j} + \frac{A_{i,j}}{N^{\alpha_{i,j}}} + \frac{B_{i,j}}{|D_i|+|D_j|^{\beta_{i,j}}}$ | Performance Prediction | 5 | 14 |
| Kaplan et al. (2020) | $L(N,D) = \left[\left(\frac{N_c}{N}\right)^{\frac{\alpha_N}{\alpha_D}} + \frac{D_c}{D}\right]^{\alpha_D}$ | Performance Prediction | 4 | 7 |
| Ghorbani et al. (2021) | $BLEU = c_B L^{-p_B}, \hat{L}_{opt}(B) = \alpha^* B^{-(p_d+p_e)} + L_\infty, \quad \alpha^* \equiv \alpha\left(\frac{N_e(p_e+p_d)}{p_e}\right)^{p_e}\left(\frac{N_d(p_e+p_d)}{p_d}\right)^{p_d}$ | Optimal Ratio, Performance Prediction | 6 | 8 |
| Gao et al. (2023) | $R_{bom}(d) = d(\alpha_{bom} - \beta_{bom}d), R_{RL}(d) = d(\alpha_{RL} - \beta_{RL}\log d)$ | Performance Prediction | 2 | 2 |
| Hilton et al. (2023) | $I^{-\beta} = \left(\frac{N_c}{N}\right)^{\alpha_N} + \left(\frac{E_c}{E}\right)^{\alpha_E}$ | Optimal Ratio, Performance Prediction | 5 | 3 |
| Frantar et al. (2023) | $L(S,N,D) = \left(a_S(1-S)^{b_S} + c_S\right)\cdot\left(\frac{1}{N}\right)^{b_N} + \left(\frac{a_D}{D}\right)^{b_D} + c$ | Optimal Ratio, Performance Prediction | 7 | 2 |
| Prato et al. (2021) | $Err(C) = Err_\infty + bC^\alpha$ | Performance Prediction | 3 | 12 |
| Covert et al. (2024) | $\log|\psi_k(z)| \approx \log|c(z)| - \alpha(z)\log(k)$ | Performance Prediction | 2 | Many |
| Hernandez et al. (2021) | $L \approx \left[\left(\frac{N_c}{N}\right)^{\frac{\alpha_N}{\alpha_D}} + \frac{D_c}{k(D_F)^\alpha(N)^\beta}\right]^{\alpha_D}$ | Performance Prediction | 3 | 1 |
| Ivgi et al. (2022) | NS | Performance Prediction | NA | NA |
| Tay et al. (2022) | None | Scaling trend | NA | NA |
| Tao et al. (2024) | $N_v^{opt} = N_v^0 * \left(\frac{N_{nv}}{N_{nv}^0}\right)^\gamma, \mathcal{L}_u = -E + \frac{A_1}{N_{nv}^{\alpha_1}} + \frac{A_2}{N_v^{\alpha_2}} + \frac{B}{D^\beta}$ | Optimal Ratio, Performance Prediction | 7 | 2 |
| Jones (2021) | plateau $= m_{boardsize}^{plateau}\cdot boardsize + c^{plateau}$ incline $= m_{boardsize}^{incline}\cdot boardsize + m_{flops}^{incline}\cdot\log flop + c^{incline}$ elo $=$ incline.clamp ( plateau , 0) | Performance Prediction | 5 | 1 |
| Zhai et al. (2022) | $E = \alpha + \beta(C+\gamma)^{-\alpha}$ | Performance Prediction | 4 | 3 |
| Dettmers & Zettlemoyer (2023) | $N^*(C) = AC^\alpha$ | Scaling trend | NA | NA |
| Dubey et al. (2024) | $N^*(C) = AC^\alpha$ | Optimal Ratio | 2 | 2 |
| Hoffmann et al. (2022) | A3: $L(N,D) = E + \frac{A}{N^\alpha} + \frac{B}{D^\beta}$ | Optimal Ratio, Performance Prediction | 5 | 3 |
| Ardalani et al. (2022) | $(\alpha x^{-\beta} + \gamma)$ | Performance Prediction | 3 | 3 |
| Clark et al. (2022) | $\log L(N,E) \triangleq +a\log N + b\log E + c\log N\log E + d$ | Performance Prediction | 4 | 3 |

Table 6: Details on power law for each paper surveyed.

| Paper | Training Runs / Law | Max. Training Flops | Max. Training Params | Max. Training Data | Data Described? | Hyperparameters Described? | How Are Model Params Counted (E.G., W/ Or W/Out Embeddings) |
|---|---|---|---|---|---|---|---|
| Rosenfeld et al. (2019) | 42-49 | | 0.7M-70M | 100M words / 1.2M images | Y | Y | Non-embedding |
| Mikami et al. | 7 | | ResNet-101 | 64k-1.28M images | Y | Y | NA |
| Schaeffer et al. (2023) | 4 | | $10^{11}$ | NA | Y | NA | Non-embedding |
| Sardana & Frankle (2023) | 47 | | 150M-6B | 1.5B-1.25T tokens | N | Y | NA |
| Sorscher et al. (2022) | 60 | | 86M (ViT) | 200 epochs | Y | Y | NA |
| Caballero et al. (2022) | 3-40 | NA | NS | NS | N | N | NA |
| Besiroglu et al. (2024) | NA | NA | 56M | | Y | NA | Non-embedding |
| Gordon et al. (2021) | 45-55 | | 56M | 28.3M-51.1M examples | Y | Y | Non-embedding |
| Bansal et al. (2022) | 10 | | 170M-800M | 500K-512M sentences (28B tokens) | Y | Y | NS |
| Hestness et al. (2017) | 9 | | upto 193M | $2^{19} - 2^{28}$ tokens, upto $2^9$ images, 2k audio hours | Y | Y | NS |
| Bi et al. (2024) | 80 | $1e17 - 3e20$ | | | Y | Y | Non-embedding |
| Bahri et al. (2024) | 8-27 | | 36.5M | upto 78k steps; 100 epochs | Y | Y | NS |
| Geiping et al. (2022) | 13 | | ResNet-18 | upto 7.6M images | Y | Y | NS |
| Poli et al. (2024) | 500 total | 8.00E+19 | 70M-7B | 400M-120B tokens | Y | Y | Non-embedding |
| Hu et al. (2024) | 36 | | 40M-2B | | Y | Y | NA |
| Hashimoto (2021) | | | | upto 600k sentences | Y | Y | NA |
| Ruan et al. (2024) | 27* -77* | | 70B-180B | 3T-6T tokens | N/A | N/A | N.S. |
| Anil et al. (2023) | 12 | 1.00E+22 | 15B | 4.00E+11 | N | N | Non-embedding |
| Pearce & Song (2024a) | 20 (simulated), 25 (real) | | 1.5B (simulated), 4.6M (real) | 23B (simulated), 500M (real) tokens | Y | Y | w/ Embedding and Non-embedding considered separately |
| Cherti et al. (2023) | 3* - 29 | | 214M | 34B (pretrain), 2B (finetune) examples | Y | Y | N.S |
| Porian et al. (2024) | 16 | 2.00E+19 | 901M | 1e6-1e10 ex / 3e11 tokens | Y | Y | w/ Embedding and Non-embedding considered separately |
| Alabdulmohsin et al. (2022) | 1* | | 110M-1B | | Mixed | Y | NA |
| Gao et al. (2024) | N.S | N.S | N.S | N.S | N | N | N.S |
| Muennighoff et al. (2024) | 142 | | 8,7B | 900B tokens | Y | Y | w/ embedding |
| Rae et al. (2021) | 4 | 6.31E+23 | 280B | | Y | Y | Non-embedding |
| Shin et al. (2023) | 17 | 0.1 PF Days | 160M | 500M-50B tokens | Y | Y | NS |
| Hernandez et al. (2022) | 56 | | 1.5M-800M | 100B tokens | N | N | NS |
| Filipovich et al. (2022) | 4 | | 57-509M | 30B token | Y | N | NS |
| Neumann & Gros (2022) | 14 | | $5 * 10^5$ | $10^4$ steps | Y | Y | NS |
| Droppo & Elibol (2021) | 5-21 | | $10^7$ | 134-23k hrs speech | Y | Y | NS |
| Henighan et al. (2020) | 6-10 | | $10^{11}$ | $10^{12}$ tokens | Y | Y | Non-embedding |
| Goyal et al. (2024) | 5 | | CLIP L/14 - 300M +63M | 32-640M samples | Y | Y | Embedding |
| Aghajanyan et al. (2023) | 21 | | 8M-6.7B | 5-100B tokens | Y | Y | Non-embedding |
| Kaplan et al. (2020) | 40-150 | | 1.5B | 23B tokens | Y | Y | Non-embedding |
| Ghorbani et al. (2021) | 12-14 | | 191-3B | NS | Y | Y | NS |
| Gao et al. (2023) | 9 | | 3B | 120-90k | N | Y | NS |
| Hilton et al. (2023) | NS | $10^{20}$ | | | Y | Y | NS |
| Frantar et al. (2023) | 48 and 112 | | 0.66M-85M | 1.8B images, 65B tokens | Y | Y | Non-embedding |
| Prato et al. (2021) | 5 | | | $10^6$ samples | Y | N | NA |
| Covert et al. (2024) | 10 | | NA | 1000 samples for IMDB | Y | Y | NA |
| Hernandez et al. (2021) | NS | $10^{21}$ | $10^8$ | | Y | N | Non-embedding |
| Ivgi et al. (2022) | 5-8 | | $10^4 - 10^8$ | varies; 500k steps PT | Y | Y | Non-embedding |
| Tay et al. (2022) | | | 16-30B | $2^{19}$ | Y | Y | NA |
| Tao et al. (2024) | 60 | | 33M-1.13B NV + 4-96k V | 4.3B-509B Characters | Y | Y | Embedding and Non-embedding considered separately |
| Jones (2021) | 200 | 1E+12-1E+17 | 4E+08-2E+09 | | Y | Y | NA |
| Zhai et al. (2022) | 44 | | 5.4M-1.8B | 1-13M images | Y | Y | NA |
| Dettmers & Zettlemoyer (2023) | 4 | | 19M-176B | NA | NA | Y | NA |
| Dubey et al. (2024) | NS | $6 * 10^{18} - 10^{22}$ | 40M-16B | | Y* | | NS |
| Hoffmann et al. (2022) | 200-450 | $6 * 10^{18} - 3 * 10^{21}$ | 16B | 5B-400B tokens | Y | Y | Non-embedding |
| Ardalani et al. (2022) | NS | $10^2$-$10^6$ TFlops | | 5M-5B samples | N | N | All are considered |
| Clark et al. (2022) | 56 | | 15M-1.3B | 130B tokens | Y | Y | Non-embedding |

Table 7: Details on training setup for each paper surveyed.

| Paper | Data Points Per Law? | Scaling Law Metric | Modification Of Final Metric? | Subsets Of Data Used |
|---|---|---|---|---|
| Rosenfeld et al. (2019) | 42-49 | Loss / Top1 Error | N | N |
| Mikami et al. | 7 | Error Rate | N | N |
| Schaeffer et al. (2023) | NA | Various downstream | NA | NA |
| Sardana & Frankle (2023) | NS | Loss | NS | NS |
| Sorscher et al. (2022) | 60 | Error Rate | NA | NA |
| Caballero et al. (2022) | 3-40 | FID, Loss, Error Rate, Elo Score | N | NS |
| Besiroglu et al. (2024) | 245 | Loss | N | N |
| Gordon et al. (2021) | 45-55 | Loss | N | N |
| Bansal et al. (2022) | NS | Loss, BLEU | NS | NS |
| Hestness et al. (2017) | NS | Token Error, CER, Error Rate, Loss | Median min. validation error across multiple training runs with separate random seeds | NS |
| Bi et al. (2024) | upto 80 | Validation bits-per-byte | NS | NS |
| Bahri et al. (2024) | upto 100 | Loss | NS | NS |
| Geiping et al. (2022) | 50 | Effective Extra Samples | Interpolation | NS |
| Poli et al. (2024) | NS | Loss | NS | NS |
| Hu et al. (2024) | NS | Loss | NS | NS |
| Hashimoto (2021) | NS | Loss | NS | NS |
| Ruan et al. (2024) | | Various downstream | N | N |
| Anil et al. (2023) | 12 | Loss | N | N |
| Pearce & Song (2024a) | 20, 5 | Loss | N | N |
| Cherti et al. (2023) | 3-29 | Error Rate | N | N |
| Porian et al. (2024) | 12 | Loss | N | N |
| Alabdulmohsin et al. (2022) | N.S. | Loss / Accuracy | N | N/A |
| Gao et al. (2024) | N.S | MSE | N.S | N.S |
| Muennighoff et al. (2024) | 142 | Loss | N | Outliers removed |
| Rae et al. (2021) | 4 | Loss | N/A | N/A |
| Shin et al. (2023) | NA | Loss | NA | NA |
| Hernandez et al. (2022) | NS | Loss | N | N |
| Filipovich et al. (2022) | NS | Loss | N | N |
| Neumann & Gros (2022) | 238 | Elo Score | N | N |
| Droppo & Elibol (2021) | NS | Loss | N | N |
| Henighan et al. (2020) | NS | Loss, Error Rate | NS | Drop smaller models |
| Goyal et al. (2024) | NS | Error Rate | N | N |
| Aghajanyan et al. (2023) | NS | Perplexity | N | N |
| Kaplan et al. (2020) | NS | Loss | NS | NS |
| Ghorbani et al. (2021) | NS | Loss, BLEU | Median of last 50k steps | NS |
| Gao et al. (2023) | 90 | RM Score | NS | NS |
| Hilton et al. (2023) | NS | Intrinsic Performance | Smoothing learning curve | Exclude early checkpoints |
| Frantar et al. (2023) | 48 and 112 | Loss | NS | NS |
| Prato et al. (2021) | 5 | Error Rate | NS | NS |
| Covert et al. (2024) | (1000-5000 )*10 | Expectation | NS | N |
| Hernandez et al. (2021) | 40-120 | Loss | NS | NS |
| Ivgi et al. (2022) | 5-8 | Loss | N | [2.5, 97.5] percentile |
| Tay et al. (2022) | NA | Loss, Accuracy | NA | NA |
| Tao et al. (2024) | 20*60 | Loss | Interpolation | NS |
| Jones (2021) | 2800 | Loss | NS | NS |
| Zhai et al. (2022) | NS | Accuracy | NS | NS |
| Dettmers & Zettlemoyer (2023) | NA | Accuracy | NA | NA |
| Dubey et al. (2024) | 150 | Loss, Accuracy | NS | NS |
| Hoffmann et al. (2022) | upto 1500 | Loss | N | Lowest loss model of a FLOP count, last 15% of checkpoints |
| Ardalani et al. (2022) | 130 | Loss | NS | NS |
| Clark et al. (2022) | 26*56 | Loss | Log | NS |

Table 8: Details on data extraction for each paper surveyed.

| Paper | Curve-Fitting Method | Loss Objective | Hyperparameters Reported? | Initialization | Are Scaling Laws Validated? |
|---|---|---|---|---|---|
| Rosenfeld et al. (2019) | Least Squares Regression | Custom error term | N/A | Random | Y |
| Mikami et al. | Non-linear Least Squares in log-log space | | N/A | N/A | Y |
| Schaeffer et al. (2023) | NA | NA | NA | NA | NA |
| Sardana & Frankle (2023) | L-BFGS | Huber Loss | Y | Grid Search | N |
| Sorscher et al. (2022) | NA | NA | NA | NA | NA |
| Caballero et al. (2022) | Least Squares Regression | MSLE | N/A | Grid Search, optimize one | Y |
| Besiroglu et al. (2024) | L-BFGS | Huber Loss | Y | Grid Search | Y |
| Gordon et al. (2021) | Least Squares Regression | | N/A | N.S. | N |
| Bansal et al. (2022) | NS | NS | N | NS | N |
| Hestness et al. (2017) | NS | RMSE | N | NS | Y |
| Bi et al. (2024) | NS | NS | N | NS | Y |
| Bahri et al. (2024) | NS | NS | N | NS | N |
| Geiping et al. (2022) | Non-linear Least Squares | | NA | Non-augmented parameters | Y |
| Poli et al. (2024) | NS | NS | N | NS | N |
| Hu et al. (2024) | scipy curvefit | NS | N | NS | N |
| Hashimoto (2021) | Adagrad | Custom Loss | Y | Xavier | Y |
| Ruan et al. (2024) | Linear Least Squares | Various | N/A | N/A | Y |
| Anil et al. (2023) | Polynomial Regression (Quadratic) | N.S. | N | N.S. | Y |
| Pearce & Song (2024a) | Polynomial Least Squares | MSE on Log-loss | N/A | N/A | N |
| Cherti et al. (2023) | Linear Least Squares | MSE | N/A | N/A | Y |
| Porian et al. (2024) | Weighted Linear Regression | weighted SE on Log-loss | N/A | N/A | Y |
| Alabdulmohsin et al. (2022) | Least Squares Regression | MSE | Y | N.S. | N.S |
| Gao et al. (2024) | N.S | N.S | N.S | N.S | N.S |
| Muennighoff et al. (2024) | L-BFGS | Huber on Log-loss | Y | Grid Search, optimize all | Y |
| Rae et al. (2021) | None | None | N/A | N/A | N |
| Shin et al. (2023) | NA | NA | NA | NA | NA |
| Hernandez et al. (2022) | NS | NS | NS | NS | NS |
| Filipovich et al. (2022) | NS | NS | NS | NS | NS |
| Neumann & Gros (2022) | NS | NS | NS | NS | NS |
| Droppo & Elibol (2021) | NS | NS | NS | NS | NS |
| Henighan et al. (2020) | NS | NS | NS | NS | NS |
| Goyal et al. (2024) | Grid Search | L2 error | Y | NA | Y |
| Aghajanyan et al. (2023) | L-BFGS | Huber on Log-loss | Y | Grid Search, optimize all | Y |
| Kaplan et al. (2020) | NS | NS | NS | NS | N |
| Ghorbani et al. (2021) | Trust Region Reflective algorithm, Least Squares | Soft-L1 Loss | Y | Fixed | Y |
| Gao et al. (2023) | NS | NS | NS | NS | Y |
| Hilton et al. (2023) | CMA-ES+Linear Regression | L2 log loss | Y | Fixed | Y |
| Frantar et al. (2023) | BFGS | Huber on Log-loss | Y | N Random Trials | Y |
| Prato et al. (2021) | NS | NS | NS | NS | NS |
| Covert et al. (2024) | Adam | Custom Loss | Y | NS | Y |
| Hernandez et al. (2021) | NS | NS | NS | NS | Y |
| Ivgi et al. (2022) | Linear Least Squares in Log-Log space | MSE | NA | NS | Y |
| Tay et al. (2022) | NA | NA | NA | NA | NA |
| Tao et al. (2024) | L-BFGS, Least Squares | Huber on Log-loss | Y | N Random Trials from Grid | Y |
| Jones (2021) | L-BFGS | NS | NS | NS | NS |
| Zhai et al. (2022) | NS | NS | NS | NS | NS |
| Dettmers & Zettlemoyer (2023) | NA | NA | NA | NA | NA |
| Dubey et al. (2024) | NS | NS | NS | NS | NS |
| Hoffmann et al. (2022) | L-BFGS | Huber on Log-loss | Y | Grid Search, optimize all | Y |
| Ardalani et al. (2022) | NS | NS | NS | NS | NS |
| Clark et al. (2022) | L-BFGS-B | L2 Loss | Y | Fixed | NS |

Table 9: Details on optimization for each paper surveyed.

## D    RECOMMENDATIONS

As seen in our analyses, many decisions in our checklist have a number of reasonable options, but those reasonable choices lead to a wide range of scaling law fits, and the observed variations do not follow any clear pattern. It is probable that variations would be even harder to predict when varying model architectures or other design decisions, removing the possibility of a universal set of best practices. However, it is certainly possible to determine that some scaling law fits are plausible or highly implausible, and to observe the stability of the fitting procedure. With the caveat that following any recommendations can not guarantee good scaling law fit, we can make some more concrete recommendations based on these observations:

**Scaling Law Hypothesis**

- Fitting fewer scaling law parameters at a time typically results in greater stability. In some cases, it may be beneficial to decompose the scaling law fitting problem into two separate procedures. Examples of this approach are the IsoFLOP procedure from Hoffmann et al. (2022), as well as fitting first the relation between $L$ and $C$, then finding the optimal $N$ and $D$ for a $C$, as seen in Porian et al. (2024).

**Training Setup**

- The trained models should include a wide range of input variables settings. For example, when the input variables to the scaling law are $L$, $N$, $D$, the included models should include a wide range of $N$ and $D$ values for each $C$, or equivalently, should include a wide range of $D/N$ ratios. If the included settings do not include the true optimum, the procedure will struggle to fit to the optimum.
- Sweeping for the optimal learning rate results in a less stable fit than fixing the learning rate. We hypothesize that this may be because the true optimal learning rate for each model and data budget size is not any of the options we consider, and thus, each model varies in the difference between its true and approximate optimal learning rate. This may introduce additional noise to the data. Due to resource constraints, we are unable to fully test this hypothesis, and it may not hold at significantly larger scale, but we recommend fixing the learning rate, or changing it according to, say, model size, according to a fixed formula.

**Data Collection**

- Results across tasks or datasets should not be mixed. Neither performance predictions nor optimal $D/N$ ratios are fixed across different evaluation settings for the same set of models.

**Fitting Algorithm**

- Scaling law fitting is sensitive to initialization; most known optimization methods for scaling laws are only able, in practice, to the shift parameters near to their initialization values. Thus, a dense search over initialization values is necessary. If there is a strong hypothesis guiding the choice of one specific initialization, such as a previously fit and validated scaling law, this will also limit the set of possible final scaling law parameter values.
- Different losses emphasize the contribution of errors from certain datapoints. The chosen loss should be suited to the distribution of datapoints and sources of noise.
- A simple grid search is unlikely to result in a good scaling law fit. Additionally, optimizers designed to fit linear relations may make assumptions about the distribution of errors and should not be used to fit a power law.

### D.1    EXAMPLE CHECKLIST

We provide one possible set of responses to our checklist, reflective of some recommendations enumerated above, and loosely based on Hoffmann et al. (2022). These answers roughly correspond to a subset of the experiments we run in §7.

---

(MIS)FITTING: SCALING LAW REPRODUCIBILITY CHECKLIST

**Scaling Law Hypothesis (§3)**

- What is the form of the power law? $L(N, D) = E + \frac{A}{N^\alpha} + \frac{B}{D^\beta}$

- What are the variables related by (included in) the power law? *N: the number of model parameters, D: the number of data tokens, and L: the model's validation loss*

- What are the parameters to fit? *A, B, E, $\alpha$, $\beta$*

- On what principles is this form derived? *This is taken from Hoffmann et al. (2022), who hypothesize this form on the basis of prior work in risk decomposition.*

- Does this form make assumptions about how the variables are related? *This form inherently assumes that N and D do not have any interaction in their effect on the scaling of L. For some experiments, we use the assumption $\alpha = \beta$ to simplify optimization.*

**Training Setup (§4)**

- How many models are trained? *Refer to Table 10*

- At which sizes? *Refer to Table 10*

- On how much data each? On what data? Is any data repeated within the training for a model? *Refer to Table 10.*

- How are model size, dataset size, and compute budget size counted? For example, how are parameters of the model counted? Are any parameters excluded (e.g., embedding layers)? How is compute cost calculated? *We include the results including and excluding embedding layers for both the total parameter count N and the total FLOP count C. We also include, for comparison, results using the estimate $C = 6ND$.*

- Are code/code snippets provided for calculating these variables if applicable? `https://github.com/hadasah/scaling_laws`

- How are hyperparameters chosen (e.g., optimizer, learning rate schedule, batch size)? Do they change with scale? *Most hyperparameters are chosen based on best practices in current literature; several are taken directly from the settings in Hoffmann et al. (2022). For learning rate, we conduct an extensive hyperparameter search across 2-3 orders of magnitude, multiplying by 2-2.5, and then conduct training at 3 learning rates, including the optimum, for nearly all (N, D) configurations.*

- What other settings must be decided (e.g., model width vs. depth)? Do they change with scale? *Refer to Table 10*

- Is the training code open source? *Yes*

| Model size | # layers | Hidden Dim | # Attn Heads | # Steps |
|---|---|---|---|---|
| 12M | 5 | 448 | 7 | {100, 200, 250, 360, 500, 750, 1,000, 4,000} |
| 17M | 7 | 448 | 7 | {100, 200, 250, 500, 750, 1,000, 1,250, 1,500} |
| 25M | 8 | 512 | 8 | {250, 360, 500, 750, 1,000, 1,500, 2,000, 8,000, 16,000} |
| 35M | 9 | 576 | 9 | {200, 250, 360, 500, 750, 1,000, 1,250, 1,500, 4,000, 16,000} |
| 50M | 10 | 640 | 10 | {250, 500, 750, 1,000, 1,250, 1,500, 1,800, 2,000, 2,500, 4,000, 16,000} |
| 70M | 12 | 704 | 11 | {250, 500, 750, 1,000, 1,250, 1,500, 2,000, 2,500, 3,000, 5,000} |
| 100M | 14 | 768 | 12 | {250, 500, 1,000, 1,500, 2,000, 2,500, 3,000, 4,000, 6,000, 12,000} |
| 200M | 18 | 960 | 15 | {500, 1,500, 6,000, 9,000} |
| 300M | 19 | 1152 | 18 | {4,000} |
| 400M | 20 | 1280 | 20 | {3,000, 6,000} |
| 1B | 26 | 1792 | 28 | {20,000} |

Table 10: Architecture and Data budget details for models trained in §7

**Data Collection(§5)**

- Are the model checkpoints provided openly? *Yes, at*
  `https://github.com/hadasah/scaling_laws`

- How many checkpoints per model are evaluated to fit each scaling law? Which ones, if so?
  *Unless clearly denoted otherwise, one checkpoint per model is evaluated; the last checkpoint. By default, no mid-training checkpoints are used, i.e., from before the termination of the cosine learning rate schedules.*

- What evaluation metric is used? On what dataset? *We use cross-entropy loss, measured on a held-out validation subset of the Common Crawl (Raffel et al., 2020) dataset.*

- Are the raw evaluation metrics modified? Some examples include loss interpolation, centering around a mean or scaling logarithmically. *No.*

- If the above is done, is code for modifying the metric provided? *Yes.*

**Fitting Algorithm (§6)**

- What objective (loss) is used? *We try various loss objectives (1) log-Huber loss, (2) MSE, (3) MAE and (4) Huber loss*

- What algorithm is used to fit the equation? *Mainly L-BFGS, but we also experiment with BFGS, non-linear least squares and grid search.*

- What hyperparameters are used for this algorithm? *Thresholds of $\{1e-4, 1e-6, 1e-8\}$, exact gradient*

- How is this algorithm initialized? *We initialize with 4500 initializations similar to Hoffmann et al. (2022).*

- Are all datapoints collected used to fit the equations? For example, are any outliers dropped? Are portions of the datapoints used to fit different equations? *No outliers are dropped in general, but we do show some results on specific subsets of models. For example, we compare the result of a scaling law fit when using only models trained at a peak learning rate of $1e-4$ or $4e-4$.*

- How is the correctness of the scaling law considered? Extrapolation, Confidence Intervals, Goodness of Fit? *Currently, we do not evaluate the correctness beyond comparing to results in literature (Hoffmann et al., 2022; Kaplan et al., 2020).*

# E    FULL ANALYSIS PLOTS

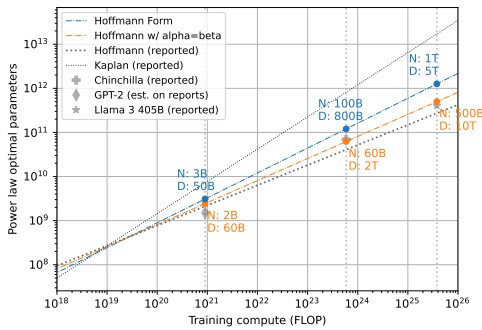

Hoffmann et al. (2022); Besiroglu et al. (2024)

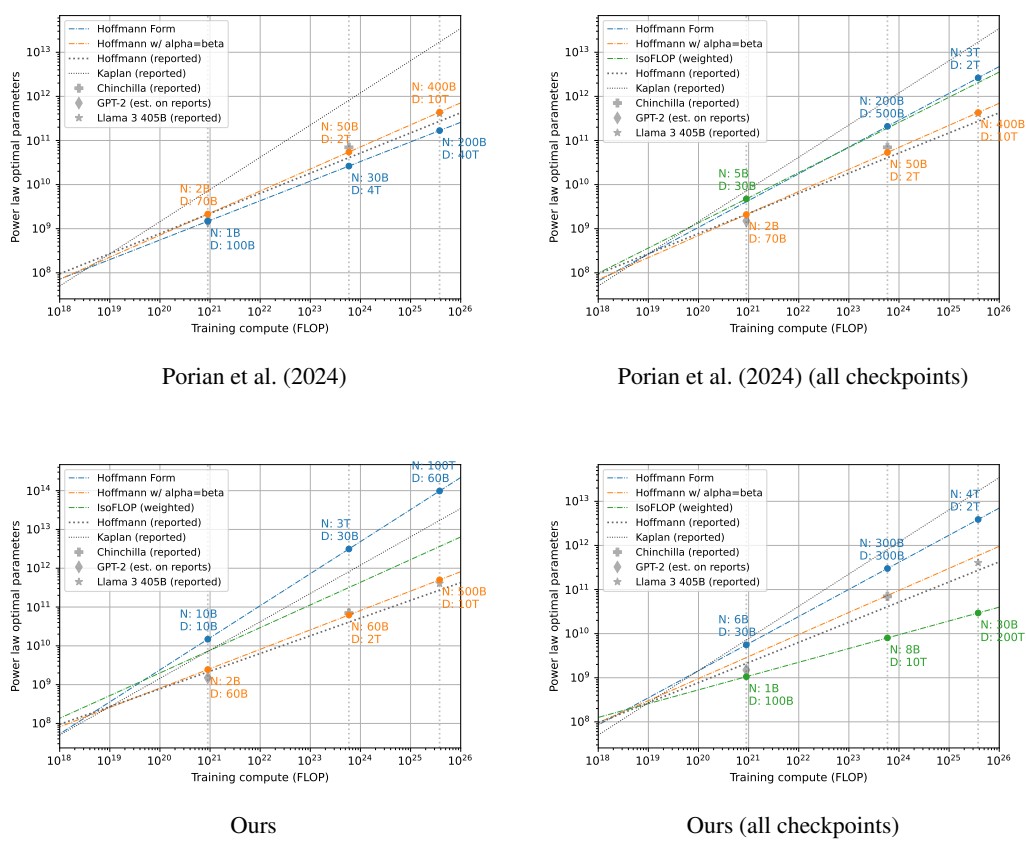

Porian et al. (2024)          Porian et al. (2024) (all checkpoints)

Ours          Ours (all checkpoints)

(a) **§3, §7.1** Using data from both Besiroglu et al. (2024) (left) and our own models (right), we compare the effects of fitting to the power law form used in Approach 3 of Hoffmann et al. (2022) with the variant used by Muennighoff et al. (2024), which assumes that the exponents $\alpha, \beta$ are equal – equivalently, that $N^*(C)$ and $D^*(C)$ scale about linearly with each other. When using only the performance of final checkpoints from both Besiroglu et al. (2024) and our own experiments, taking this assertion results in a law much closer to the one reported by Hoffmann et al. (2022). On our own models, we also show results when using the IsoFLOP approach from Hoffmann et al. (2022). As we are using only results from final model checkpoints, the size of the data input to the IsoFLOP approach in this case is reduced.

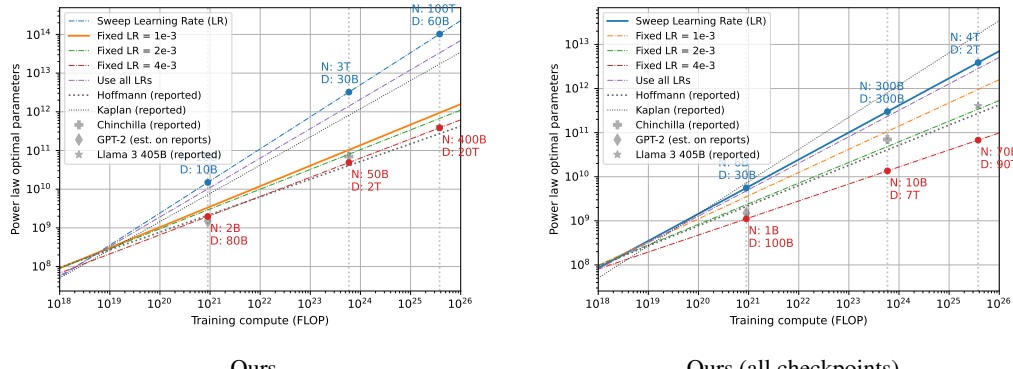

Ours                                    Ours (all checkpoints)

(b) **§4, §7.2** With our models, we simulate the effects of not sweeping the learning rate. As a baseline, (1) we sweep at each $(N, D)$ pair for the optimal learning rate over a range of values at most a multiple of 2 apart. Next, (2) we use a learning rate of 1e-3 for all $N$, the optimal for our 1 billion parameter models, and do the same for (3) 2e-3 and (4) 4e-3, which is optimal for our 12 million parameter models. Lastly, we use all models across all learning rates at the same $N$ and $D$. Results vary dramatically across these settings. Somewhat surprisingly, using all learning rates results in a very similar power law to sweeping the learning rate, whereas using a fixed learning rate of 1e-3 or 4e-3 yields the lowest optimization loss or closest match to the Hoffmann et al. (2022) power laws, respectively.

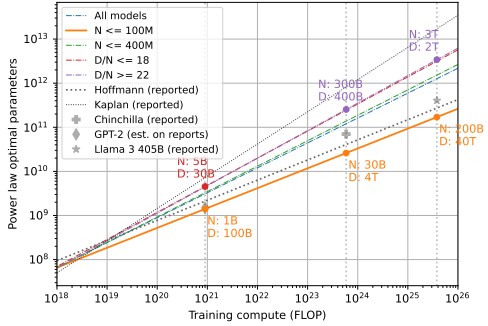

Hoffmann et al. (2022); Besiroglu et al. (2024)

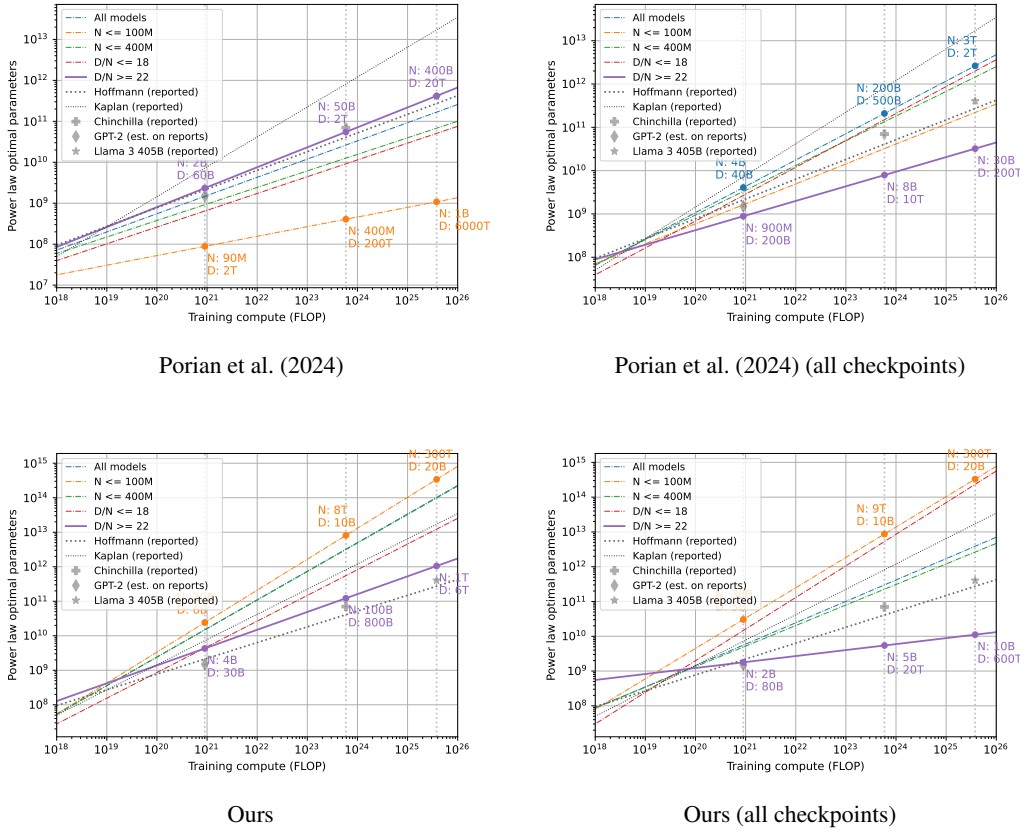

Porian et al. (2024)                    Porian et al. (2024) (all checkpoints)

Ours                                      Ours (all checkpoints)

(c) **§4, §7.2** From all 3 datasets, we choose subsets with $(N, D)$ values which fit a particular method one might have of setting up training. We fit with (1) all models, which for our dataset, ranges from 12 million to 1 billion parameters, then with (2) only models of up to about 100 million parameters or (3) up to 400 million parameters. We also compare the effects of a higher or lower hypothesis about the optimal $D/N$ ratio, including (4) only models with $D/N \leq 18$ or (5) $D/N \geq 22$. These ranges are designed to exclude $D/N = 20$, the rule of thumb based on Hoffmann et al. (2022). The minimum or maximum $D/N$ ratio tested does skew results; above $10^{22}$ FLOPs, (4) and (5) fit to optimal ratios $D/N < 18$ and $D/N > 22$, respectively. Removing our largest models in (2) also creates a major shift in the predicted optimal $D/N$.

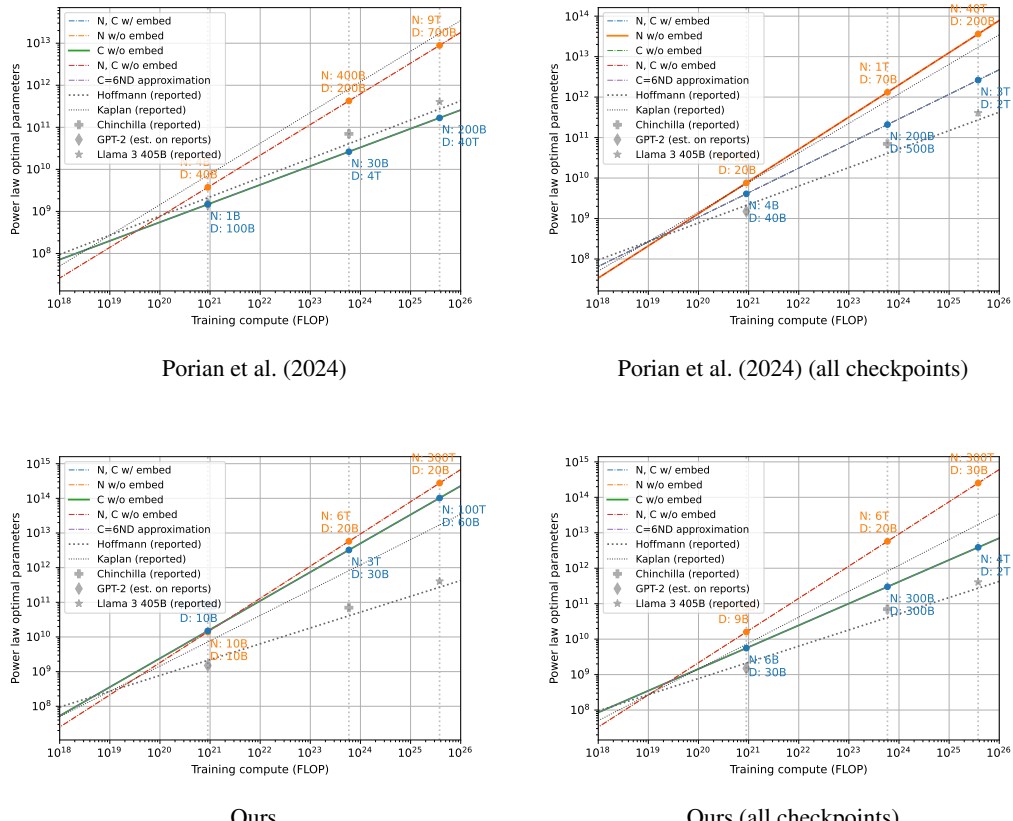

(d) **§4, §7.2** With our own data and those from Porian et al. (2024), we fit power laws to the same sets of models, while varying the ways we count $N$ and $C$. We compare (1) including embeddings, which is our baseline, with (2) excluding embeddings only in $N$, (3) excluding embeddings only in $C$, (4) excluding embeddings in both $N$ and $C$. We also compare to using the $C = 6ND$ approximation, including embedding parameters. Throughout this work, we calculate the FLOPs in a manner similar to Hoffmann et al. (2022), and we open source the code for these calculations. With both datasets, the exclusion of embeddings in FLOPs has very little impact on the final fit. Similarly, using the $C = 6ND$ approximation has no visible impact. For the Porian et al. (2024) models, the exclusion of embedding parameters in the calculation of $N$ results in scaling laws which differ substantially, and with increasing divergences at large scales.

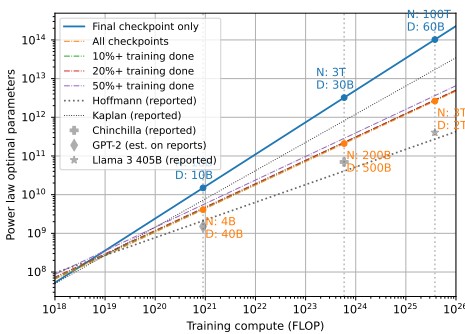

Porian et al. (2024) (all checkpoints)

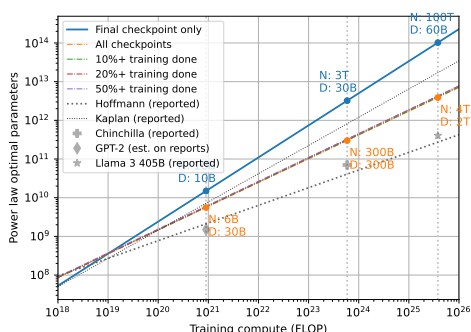

Ours (all checkpoints)

(e) **§5, §7.3** With models from Porian et al. (2024) and our own dataset, we compare (1) fitting a power law using only the final checkpoint with (2) using all mid-training checkpoints (3) using all checkpoints, starting 10% through training, (4) the same, starting 20% through training, and (5) the same again, starting 50% through training. We observe that (2)-(5) consistently yields power laws more similar to that reported by Hoffmann et al. (2022), so we also repeat all analyses in Figures 2a-2d. Using mid-training checkpoints sometimes results in more stable fits which are similar to the Hoffmann et al. (2022) scaling laws, but the effect is noisy and dependent on other decisions

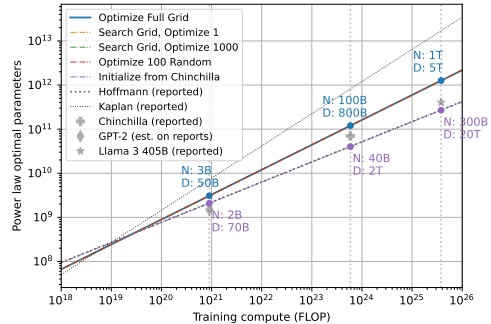

Hoffmann et al. (2022); Besiroglu et al. (2024)

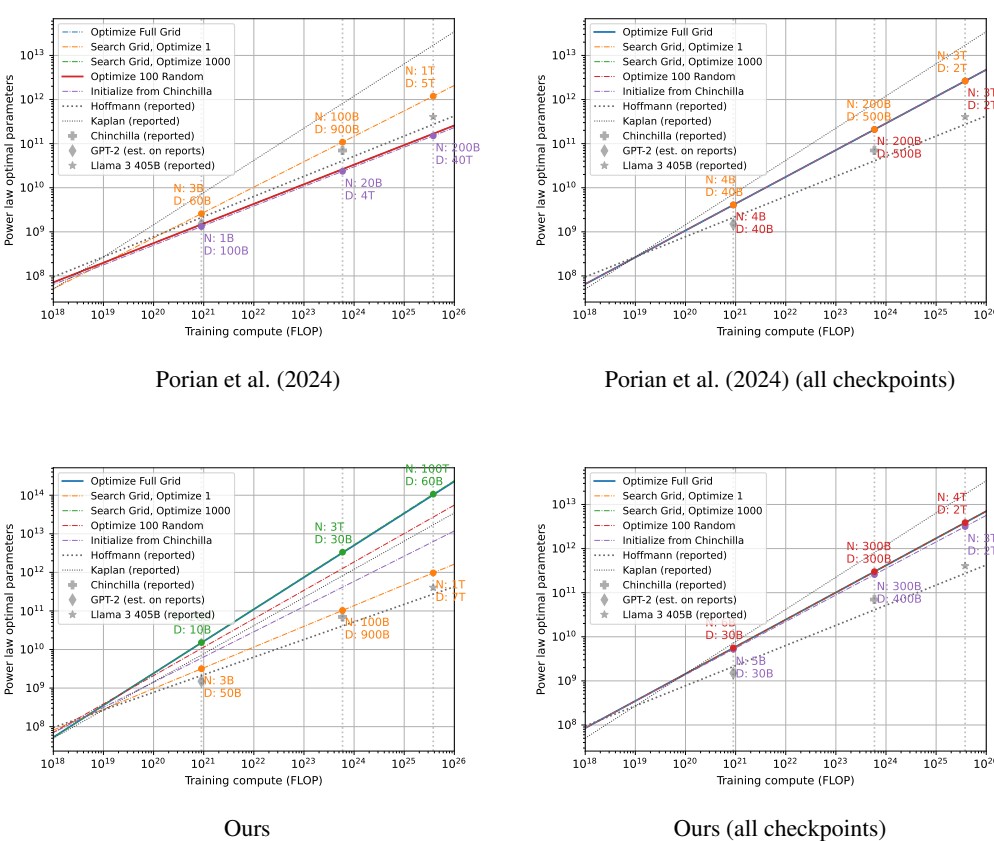

Porian et al. (2024)  Porian et al. (2024) (all checkpoints)

Ours  Ours (all checkpoints)

(f) **§6, §7.4** We fit to data from all 3 datasets to experiment with the initialization of parameters in the power law. We start with (1) optimizing every point in a grid search of 6x6x5x5x5=4500 initializations (Hoffmann et al., 2022), (2) randomly sampling from only a single initialization in this grid, (3) searching for the lowest loss initialization point (Caballero et al., 2022), (4) randomly sampling 100 points, and (5) initializing with the coefficients found in Hoffmann et al. (2022), as Besiroglu et al. (2024) does. With the Besiroglu et al. (2024) data, (5) yields a fit nearly identical to that reported by Hoffmann et al. (2022), although (1) results in the lowest fitting loss. With the Porian et al. (2024) data, all approaches except (2) yield a very similar fit, which gives a recommended $D/N$ ratio similar to that ofHoffmann et al. (2022). However, using our data, (2) optimizing over only the most optimal initialization yields the best match to the Hoffmann et al. (2022) power laws, followed by (5) initialization from the reported Hoffmann et al. (2022) scaling law parameters. Optimizing over the full grid yields the power law which diverges most from the Hoffmann et al. (2022) law, suggesting the difficulty of optimizing over this space, and the presence of many local minima.

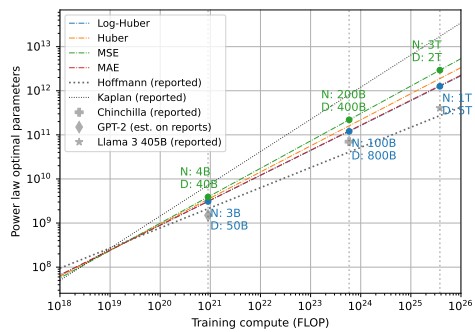

Hoffmann et al. (2022); Besiroglu et al. (2024)

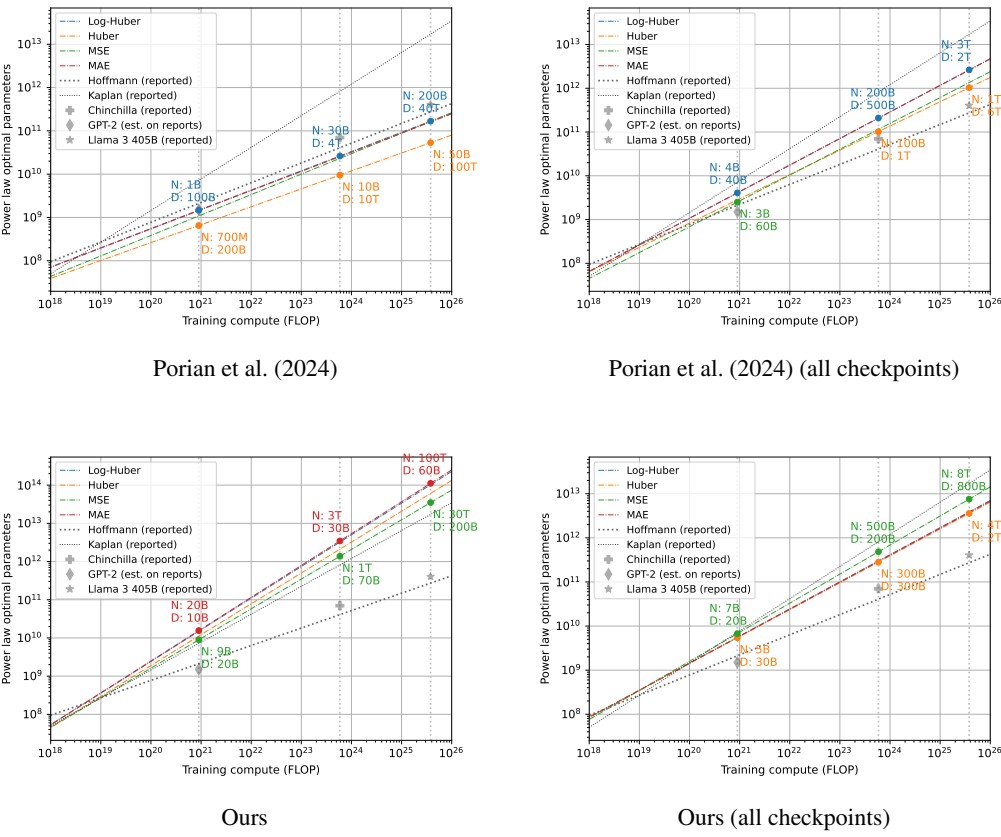

Porian et al. (2024)

Porian et al. (2024) (all checkpoints)

Ours

Ours (all checkpoints)

(g) **§6, §7.4** We fit a power law to data from from all 3 datasets, minimizing different objective functions: (1) the baseline log-Huber loss, (2) MSE, (3) MAE, and (4) the Huber loss. The resulting power laws are less disparate than when varying many of the other factors discussed above and generally fall near the power law parameters reported by Kaplan et al. (2020) and Hoffmann et al. (2022), but this is still a wide range of recommended optimal parameter counts for each compute budget. Overall, the loss function behavior is not predictable, given the differences between loss functions when looking at the power laws resulting from these three sources of data.

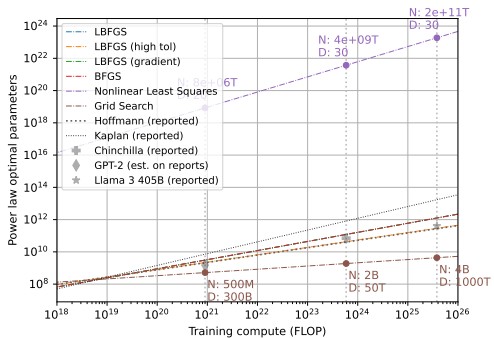

Hoffmann et al. (2022); Besiroglu et al. (2024)

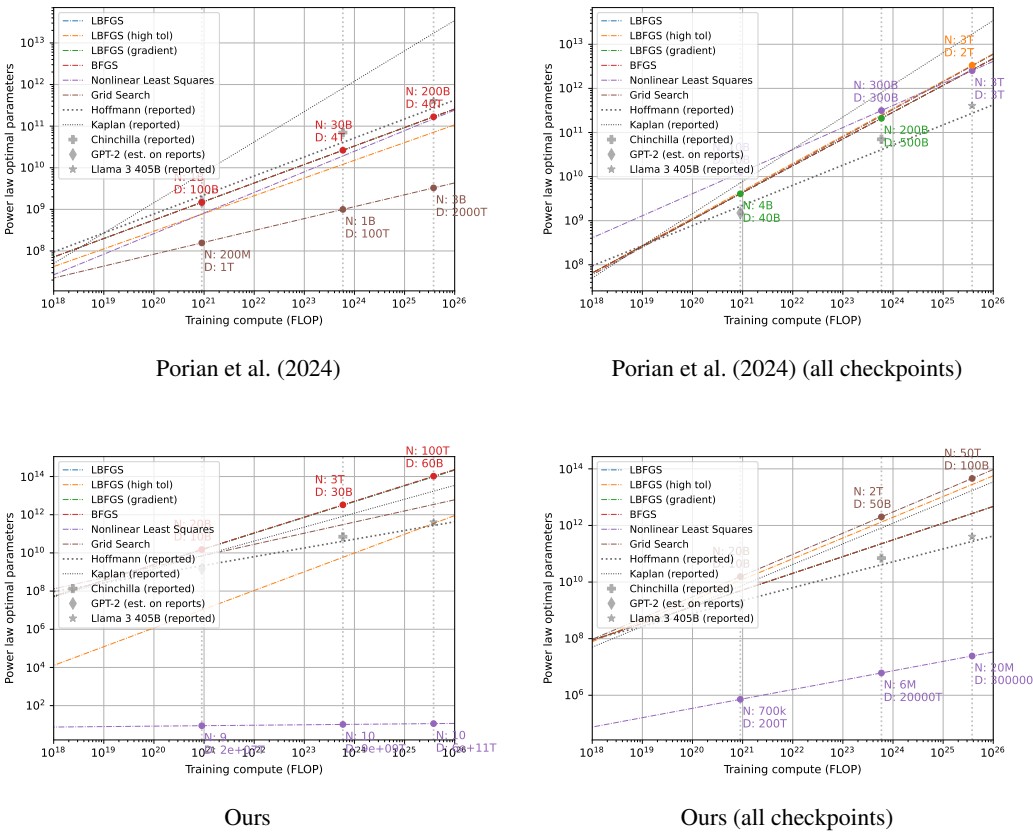

Porian et al. (2024)

Porian et al. (2024) (all checkpoints)

Ours

Ours (all checkpoints)

(h) **§6, §7.4** We fit a power law to data from all 3 datasets using various optimizers, beginning with the original (1) L-BFGS. L-BFGS and BFGS implementations have an early stopping mechanism, which conditions on the stability of the solution between optimization steps. We set this threshold for L-BFGS to a (2) higher value 1e-4 (stopping earlier). We found that using any higher or lower values resulted in the same solutions or instability for all three datasets. L-BFGS and BFGS also have the option to use the true gradient of the loss, instead of an estimate, which is the default. In this figure, we include this setting, (3) using the true gradient for L-BFGS. We compare these L-BFGS settings to (4) BFGS. We test the same tolerance value and gradient settings, and find that none of these options change the outcome of BFGS in our analysis, and omit them from this figure for legibility. Finally, we compare to (5) non-linear least squares and (6) pure grid search, using a grid 5 times more dense along each axis as we used initialization with other optimizers. This density is chosen to approximately match the runtime of L-BFGS. Many of these optimizers do converge to similar solutions, but this depends on the data, and the settings which diverge from this majority do not follow any immediately evident pattern.

Figure 2: (**§7**) We study the effects of various decisions in the fitting of a power law, as outlined in our checklist (Appendix B) and detailed in §3-§6. For each set of analyses, we the scaling laws found by (Kaplan et al., 2020) and (Hoffmann et al., 2022) for comparison. We also include markers indicating 3 existing models for comparison purposes: Llama 3 405B (Dubey et al., 2024), the Chinchilla model (Hoffmann et al., 2022), and an estimate of the 1.5B GPT-2 model (Radford et al., 2019), for which we know details of the dataset storage size and word count, but not an exact count of data BPE tokens, which we estimate at 100B. We additionally annotate, at the compute budget $C$ for each of these 3 reference points, the maximum and minimum *predicted* (i.e. extrapolated) optimal model parameter count $N_{opt}$ and data budget $D_{opt}$ from the fitted power laws. We use a thicker, solid line for the method in each plot which achieves the lowest optimization loss, with the exception of the plots comparing power law form, those comparing loss functions and those comparing optimizers, for which this would be nonsensical. We find overall, throughout our analyses, that all of the decisions we explore have an impact on the final fit of the power law, supporting our conclusion that more thorough reporting of these decisions is critical for scaling law reproducibility.

