# OpenReview forum: "(Mis)Fitting Scaling Laws: A Survey of Scaling Law Fitting Techniques in Deep Learning"
_ICLR.cc/2025/Conference — ICLR 2025 Poster_

### Official Review · Reviewer_pBpK · 2024-10-25

**Soundness:** 3
**Presentation:** 2
**Contribution:** 2
**Rating:** 5
**Confidence:** 3

**Summary:**

The paper presents a survey on the fitting of scaling laws, and argue that current practices are lacking in scientific rigor. Apart from an extensive survey, the authors presents a reproducibility checklist, and compare 51 papers to this checklist. They generally find that important details are underreported, e.g. the method to calculate model parameters might not be given. They also provide a replication study of Hoffman, using data extracted from the paper PDF and data they’ve collected themselves. Here they find that subtle choices in the curve fitting can result in significantly different conclusions.

**Strengths:**

- Scaling laws is an important topic, and scientific rigor here can benefit the research community.
- The authors illustrate how subtle choices in the curve fitting can cause significant results
- Section 7 is great.

**Weaknesses:**

- Significant parts of the papers are dedicated to a survey. I’m not sure survey papers are the right fit for ICLR main track.
- There are not so many empirical results.

**Questions:**

- could you provide explicit recommendation regarding how to perform the curve fitting? I think this is different from a checklist which allows reproduction.
- Could you expand section 7?

---

> ### Author Response · Authors · 2024-11-15
>
> Thank you for your constructive review and feedback - we also hope that this paper will enable researchers and practitioners to train large-scale models more effectively. We address your concerns below:
>
> - *Recommended action:* We choose to avoid being overly prescriptive in our recommendations because there is no set of actions which can guarantee a good scaling law fit, nor even make a fit extremely likely.
>   (1) It is intractable to establish what the desired final scaling law is, and therefore to measure the goodness of any scaling law fit, because we don't know the ground truth of model performance at all scales. Any attempt to estimate the goodness of a scaling law fit can only consider the goodness fit at a small number of points with limited scale, and it is unclear how heavily to weigh the goodness at each point.
>
>    (2) As seen in our S7 analyses, many decisions in our checklist have a number of reasonable options, but those reasonable choices lead to a wide range of scaling law fits, and the observed variations do not follow any clear pattern. It is probable that variations would be even harder to predict when varying model architectures or other design decisions.
>
>   However, it is certainly possible to determine that some scaling law fits are plausible or highly implausible, and to observe the stability of fits. For example, in Figure 2(a), neither of the recommended data/parameters ratios of ~1/2200 or 3000/1 at 10^25 FLOPs are likely to be the true optimal settings, and the loss predictions at those points are also unlikely to be close to ground truth. Based on these observations, we can make some more concrete recommendations, with the caveat that following the recommendations is no absolute guarantee of good scaling law fit. For example, Appendix Figure 3(c) suggests the importance of using a large range of data/parameter ratios and absolute model parameter counts across the experiments, since using too narrow a range can skew the final fit dramatically. Additionally, no paper we know of has achieved a plausible scaling law fit by directly optimizing a power law form for performance prediction with 5 or more scaling law parameters, so we recommend considering an IsoFLOP or multi-stage approach (e.g., fitting a relation between L and C, then a relation between C and optimal N). We will update the camera ready with these recommendations.
>
> - *Section 7:* We have expanded results in Figure 3 (Appendix). We will also update the camera ready version to include additional analyses, as well as models up to 1B parameters (not included in the submission due to resource constraints). It is unfortunately impossible to enumerate and execute all possible variations on scaling laws, but we intend to add as many as possible given space constraints. We would be eager to hear any specific suggestions the reviewer might have.
>
> - *Surveys:* While surveys and position papers at ML conferences are rare, we believe that the ICLR audience is the correct audience for this work. Given our contributions of (1) elucidating and organizing the factors influencing scaling law studies, (2) empirically analyzing several of these factors and (3) providing a checklist to help both researchers and readers of scaling law papers, we hope that there is room for work like ours at ICLR.
>
> We hope that this is helpful, and that you consider increasing your score if your concerns are addressed. We are happy to answer follow up questions.

---

> ### Author Response · Authors · 2024-11-25
> **Updates to the Draft**
>
> Dear Reviewer,
>
> Based on the discussion, we have updated the draft:
>
> - We have defined the categories of our checklist explicitly in the Appendix.
> - We have added a section on Recommendations in Appendix E, and provided an example checklist.
> - We have added to Section 4 a discussion on the maximum scales of the papers we have surveyed.
>
> If you find these updates to be sufficient, we hope that you will increase our scores accordingly. We are happy to answer any follow up questions.

---

> > ### Comment · Reviewer_pBpK · 2024-12-01
> >
> > Thanks for your reply, I will retain my score.

---

### Official Review · Reviewer_kn4j · 2024-10-29

**Soundness:** 4
**Presentation:** 4
**Contribution:** 4
**Rating:** 8
**Confidence:** 5

**Summary:**

This work revisits scaling laws and factors influencing the results reported in recent papers. The paper takes a "review approach", outlining some of the debates in recent literature as well as common strategies. To draw conclusions and emphasize their points, the last pages are dedicated to an in-depth analysis by the authors on small to moderate size transformers.

**Strengths:**

I really like this paper and think it is of great value for the community to "recap" scaling law results and provide a critical discussion, complemented by experiments showing which factors matter when choosing a scaling law. It was really a pleasant read, quality of writing is good and the motivation clear.

**Weaknesses:**

One could maybe claim the paper is not too constructive, as it shows that choices (optimizer, fitting method, lr annealing, data) matter when fitting a scaling law: there is no correct answer. However, this conclusion demystifies the topic, which I like very much: there is no magic, just common choices and "usual" results. This said, there are a couple of very minor points.

1) Proposing a checklist is helpful, but, as the authors themselves seem to hint, the number of factors to account for is potentially infinite. What about Adam beta2? What about weight decay? What about hybrid algorithms? What about qk norm and new tricks? The reality this paper points out is that, indeed, such choices matter, and I do not think any checklist can be conclusive.

2) section 7.1: why did you decide to set alpha=beta?

3) The paper lacks a bit of conclusions: what should researchers do? should we trust scaling laws? what are the things that hold true despite changing the setting? Is there some practical rule for scaling that holds approximately in your experiments? (would have been interesting alpha and beta)

typo spot: "was was" in the abstract, repetition.

**Questions:**

-

---

> ### Author Response · Authors · 2024-11-15
> **Rebuttal**
>
> Thank you for your positive review and suggestions - we hope that this paper will enable researchers and practitioners to train large-scale models more effectively. We will fix the typos flagged, and address your concerns below:
>
> - *Inherent Incompleteness of Checklists:* We agree that a checklist cannot be exhaustive or complete. We do, however, attempt to go over which factors matter as extensively as possible. This is similar to how Model Cards (Mitchell et al [1]) are also not necessarily exhaustive, but are still useful. In fact, much of the data on these papers that we were able to gather was only possible because the authors open-sourced several artifacts (Table 5 in the Appendix). Despite the impossibility of any checklist being exhaustive, we believe laying out this checklist will enable those who read and build upon scaling laws papers to approach papers with skepticism, and compare results across studies. Moreover, similar to Model Cards, we hope that this facilitates more effective scaling studies by virtue of encouraging researchers to consider each decision in study development more carefully.
> - *Section 7.1:* The trick of setting alpha=beta is taken from Muenninghoff, et. al. They do this to simplify the optimization problem, which now needs to fit only 4 scaling law parameters, instead of 5. This inherently assumes that the optimal data and model parameters scale linearly with each other – that their ratio is fixed. Hoffmann, et.al., fit a scaling law resulting in alpha ~= beta, and recommended fixing data:parameters at 20:1. We use alpha=beta as one setting in our comparisons, to understand the effects of making such an assumption (Figure 3a in the Appendix).
>
> - *Recommended action:* We choose to avoid being overly prescriptive in our recommendations because there is no set of actions which can guarantee a good scaling law fit, nor even make a fit extremely likely.
>   (1) It is intractable to establish what the desired final scaling law is, and therefore to measure the goodness of any scaling law fit, because we don't know the ground truth of model performance at all scales. Any attempt to estimate the goodness of a scaling law fit can only consider the goodness fit at a small number of points with limited scale, and it is unclear how heavily to weigh the goodness at each point.
>
>    (2) As seen in our S7 analyses, many decisions in our checklist have a number of reasonable options, but those reasonable choices lead to a wide range of scaling law fits, and the observed variations do not follow any clear pattern. It is probable that variations would be even harder to predict when varying model architectures or other design decisions.
>
>   However, it is certainly possible to determine that some scaling law fits are plausible or highly implausible, and to observe the stability of fits. For example, in Figure 2(a), neither of the recommended data/parameters ratios of ~1/2200 or 3000/1 at 10^25 FLOPs are likely to be the true optimal settings, and the loss predictions at those points are also unlikely to be close to ground truth. Based on these observations, we can make some more concrete recommendations, with the caveat that following the recommendations is no absolute guarantee of good scaling law fit. For example, Appendix Figure 3(c) suggests the importance of using a large range of data/parameter ratios and absolute model parameter counts across the experiments, since using too narrow a range can skew the final fit dramatically. Additionally, no paper we know of has achieved a plausible scaling law fit by directly optimizing a power law form for performance prediction with 5 or more scaling law parameters, so we recommend considering an IsoFLOP or multi-stage approach (e.g., fitting a relation between L and C, then a relation between C and optimal N). We will update the camera ready with these recommendations.
>
>
> We hope that this addresses your concerns, and are happy to answer follow up questions.
>
>
> [1] https://arxiv.org/abs/1810.03993

---

> > ### Comment · Reviewer_kn4j · 2024-11-23
> > **Thanks!**
> >
> > Thanks for your answer, keeping my score.

---

> > > ### Author Response · Authors · 2024-11-25
> > >
> > > Thank you for the support - based on the discussion, we have also updated the draft:
> > > - We have defined the categories of our checklist explicitly in the Appendix.
> > > - We have added a section on Recommendations in Appendix E, and provided an example checklist.
> > > - We have added to Section 4 a discussion on the maximum scales of the papers we have surveyed.

---

### Official Review · Reviewer_vpGv · 2024-11-03

**Soundness:** 3
**Presentation:** 3
**Contribution:** 2
**Rating:** 5
**Confidence:** 5

**Summary:**

The authors survey a large corpus of papers that involve scaling laws, and find that many papers underreport necessary details for reproducibility, which they demonstrate with experiments that demonstrate a large variability depending on those exact choices of details. They propose a checklist for authors to consider when publishing scaling laws.

**Strengths:**

The paper gives a good overview of many papers on scaling laws, and nicely categorizes the important steps: functional form, training setup, data(points) extraction, curve fitting. The checklist provides a clear way of reproducibility and quality assessment of scaling experiments. I think the topic of scaling law studies is important and relevant, and the writing is clear.

**Weaknesses:**

The main concern for me is the following: what is the main goal the authors are trying to convey? To me, there are two obvious takeaways, which is 1) changes to the scaling law setup can change the results drastically, and 2) previous papers very much underreport crucial details. However, both of these things are rather clear already to the community and also illustrated by published papers: for example, point 1) is shown by Porian et al., and point 2) is a broader critique of reproducibility problems, which (unfortunately) is a generic problem. I do not see a clear and actionable interpretation beyond that. For instance, how do the different choices of fitting actually affect the scaling laws? (The assessment is mostly just “the results vary dramatically” — but how?) What should I as a researcher now do for my future scaling studies, having read your paper, beyond using the checklist? Are there clearly ‘wrong’ or ‘right’ choices? Was there a most predictive scaling law (e.g. when you leave out some experiments as a validation set)?

To be clear, I very much believe there is merit in a survey or pointing out these problems; as it stands, however, the paper is foremost “just” a survey, and I am not convinced this merits publishing at the conference.

Some additional comments:
 * The paper template says ICLR 2024
 * The Figures are unfortunately of low quality (very pixelated), especially considering the fact that it’s natural to zoom in to compare the many lines and details. I suggest the authors include the pdf forms for proper rendering.

**Questions:**

I have already listed direct questions in the section above, and I would be open to discuss this in the rebuttal. I hope the authors see the comments to be constructive, and can clarify or improve the distinctive value of the paper.

---

> ### Author Response · Authors · 2024-11-15
> **Rebuttal**
>
> Thank you for your constructive review - we also hope that this paper will enable researchers and practitioners to train large-scale models more effectively. We address your concerns below:
>
> - *Comparison to Porian et al*: Porian, a work contemporary to ours, is focused more narrowly on the specific differences between Kaplan et al and Hoffman et al scaling laws - namely, FLOP counting and training setup. While our work has similar motivations, we consider more broadly the factors that may change the conclusions of a scaling study, namely (1) the form of the hypothesis itself (2) training setup (which Porian et al touch upon), (3) evaluating the trained models and (4) fitting the scaling law. We empirically show that varying several of these factors can change the final scaling law by an order of magnitude.
>
> - *Goal of Paper:* We agree that the ultimate hope is to make specific recommendations about scaling laws. As our paper has been presented, our main recommendation is for researchers who are fitting scaling laws to open source all artifacts, or, short of that, to report items on our checklist as fully as possible; and for those reading others' scaling law works, to approach such works with skepticism, closely examining each aspect of the methodology with our checklist.
>
> - *Recommended action:* We choose to avoid being overly prescriptive in our recommendations because there is no set of actions which can guarantee a good scaling law fit, nor even make a fit extremely likely.
>   (1) It is intractable to establish what the desired final scaling law is, and therefore to measure the goodness of any scaling law fit, because we don't know the ground truth of model performance at all scales. Any attempt to estimate the goodness of a scaling law fit can only consider the goodness fit at a small number of points with limited scale, and it is unclear how heavily to weigh the goodness at each point.
>
>    (2) As seen in our S7 analyses, many decisions in our checklist have a number of reasonable options, but those reasonable choices lead to a wide range of scaling law fits, and the observed variations do not follow any clear pattern. It is probable that variations would be even harder to predict when varying model architectures or other design decisions.
>
>   However, it is certainly possible to determine that some scaling law fits are plausible or highly implausible, and to observe the stability of fits. For example, in Figure 2(a), neither of the recommended data/parameters ratios of ~1/2200 or 3000/1 at 10^25 FLOPs are likely to be the true optimal settings, and the loss predictions at those points are also unlikely to be close to ground truth. Based on these observations, we can make some more concrete recommendations, with the caveat that following the recommendations is no absolute guarantee of good scaling law fit. For example, Appendix Figure 3(c) suggests the importance of using a large range of data/parameter ratios and absolute model parameter counts across the experiments, since using too narrow a range can skew the final fit dramatically. Additionally, no paper we know of has achieved a plausible scaling law fit by directly optimizing a power law form for performance prediction with 5 or more scaling law parameters, so we recommend considering an IsoFLOP or multi-stage approach (e.g., fitting a relation between L and C, then a relation between C and optimal N). We will update the camera ready with these recommendations.
>
>
> - *Surveys:* While surveys and position papers are only a small portion of ML conference submissions,  we believe that the ICLR audience is the correct audience for this work. Given our contributions of (1) elucidating and organizing the factors influencing scaling law studies, (2) empirically analyzing several of these factors and (3) providing a checklist to help both researchers and readers of scaling law papers, we hope that there is room for work like ours at ICLR.
>
> - *Additional comments:* Thank you for bringing these to our attention. We will update the draft with vector versions of the images, and update the year.
>
> We hope that our clarifications are helpful, and that you consider increasing your score if your concerns are addressed. We are happy to answer follow up questions.

---

> ### Author Response · Authors · 2024-11-25
> **Updates to the Draft**
>
> Dear Reviewer,
>
> Based on the discussion, we have updated the draft:
>
> - We have defined the categories of our checklist explicitly in the Appendix.
> - We have added a section on Recommendations in Appendix E, and provided an example checklist.
> - We have added to Section 4 a discussion on the maximum scales of the papers we have surveyed.
>
> If you find these updates to be sufficient, we hope that you will increase our scores accordingly. We are happy to answer any follow up questions.

---

> > ### Comment · Reviewer_vpGv · 2024-11-27
> >
> > I want to thank the authors for providing responses to the questions raised by me and other reviewers. Given the deliberate and nuanced discussion of these points (such as lack of clear guidelines for how to do scaling laws) both here and in the paper, I have raised my score to 5. Nonetheless I still remain uncertain that a work that is primarily a survey merits acceptance, also under the consideration that the problem of scaling law misfits is known to a large majority of the community.

---

### Official Review · Reviewer_PhzZ · 2024-11-08

**Soundness:** 2
**Presentation:** 2
**Contribution:** 3
**Rating:** 5
**Confidence:** 2

**Summary:**

This paper surveyed/ more than 50 papers about the scaling law of language models. Authors discussed different aspects of scaling law including fitting forms, model training, data extraction, and fitting optimization. Based on that, authors provided a checklist, which helps to transparent settings for reproducible results in future research. Experiments also were conducted to verify their replication and analyses.

**Strengths:**

This paper has several strengths:
- This paper considers a timely topic, the scaling law of language models. Understanding this topic will help to effectively train LLMs, avoiding resource overuse.
- Authors discussed the discrepancies in experiment settings of different papers and empirically verified it. Results are aligned with previous works.
- Authors open-sourced their code to reproduce results which benefits the community since the source code is usually absent from previous papers.

**Weaknesses:**

Despite these strengths, this paper has several weaknesses:
- The scale of the model in experiments is not big enough. Authors consider only models with less than 400M params, ignoring the existence of larger models with billions of params.
- The writing in some parts of the main paper causes confusion. E.g., Section 5 is about data extraction after training, I was confused by which kind of data could be extracted.

**Questions:**

Please see the weaknesses.

---

> ### Author Response · Authors · 2024-11-15
> **Rebuttal**
>
> Thank you for your constructive review - we also hope that this paper will enable researchers and practitioners to train large-scale models more effectively. We address your concerns below:
>
> - *Model scale:* While state-of-the-art LLMs have hundreds of billions of parameters, there is emerging evidence [1] that it is not necessary to reach billions of parameters in order to study scaling laws. We note that of the papers we survey, most do not reach a maximum model scale of more than 1B parameters - we will add a discussion of this in the next draft. Notably, we refer to Kaplan, et. all, and Hoffmann, et. al., which use maximum model scales of 1.5B and 16B, respectively, and heavily overrepresent models with fewer than 100M parameters in their scaling law data.
>
>   We are training up to 1B parameter models and will include these in the camera-ready, but were unable to include them at submission time due to resource constraints - training this model for 20B tokens costs about 1000 GPU-hours on Nvidia A40 machines, and a paper which includes this model size would inevitably need to partially or fully train several such models.
>
> - *Confusing terminology:* The particular stage of scaling law fitting that we call "data extraction" is evaluating various checkpoints of our trained models to collect data points that will be used to fit a scaling law equation in the next stage. We could call this “Data for Fitting the Scaling Law” instead although we were trying to be more succinct  - we are open to suggestions for new terminology. We will ensure that there are more precise definitions of this term (and other terms introduced) in the next draft.
>
> We hope that this is helpful, and that you consider increasing your score if your concerns are addressed. We are happy to answer follow up questions.
>
> [1] https://openreview.net/forum?id=xGM5shdGJD

---

> > ### Comment · Reviewer_PhzZ · 2024-11-25
> >
> > We thank authors for clarifying my confusion. However, my main concern, which is about the correctness of scaling law with model size, are not fully addressed during the rebuttal. I decided to keep my score unchanged.

---

> ### Author Response · Authors · 2024-11-25
> **Updates to the Draft**
>
> Dear Reviewer,
>
> Based on the discussion, we have updated the draft:
>
> - We have defined the categories of our checklist explicitly in the Appendix.
> - We have added a section on Recommendations in Appendix E, and provided an example checklist.
> - We have added to Section 4 a discussion on the maximum scales of the papers we have surveyed.
>
> If you find these updates to be sufficient, we hope that you will increase our scores accordingly. We are happy to answer any follow up questions.

---

### Comment · Area_Chair_kcve · 2024-11-30

Dear reviewers, and authors:

Thank you all for engaging with the review process.
Several reviewers were uncertain about whether surveys / meta-analyses are appropriate for ICLR.
Several points of guidance below:
* Since there is no official policy on whether surveys are in-scope for ICLR, this matter should be handled on a case-by-case basis.
* Historically, ICLR has been welcoming to papers that are valuable and of interest to the community, even if they are outside the traditional scope of ML conferences. For example, the einops paper (ICLR 2022 Oral) was not a research paper: https://openreview.net/forum?id=oapKSVM2bcj
* The current paper appears to be closer to a “meta-analysis” work than a purely-survey paper. The authors have introduced a taxonomy of the major sources of inconsistency in the literature, and have also made efforts to independently reproduce experiments from prior works.
* Thus, we encourage reviewers to score this paper based on: (1) correctness, and (2) whether the ICLR community would benefit from reading this work. Note that surveys and meta-analyses cannot be judged by the same criteria as research papers — for example, novelty of claims or experiments is less of a factor. For surveys and meta-analyses, however, it is important that they are comprehensive within their stated scope.

---

### Meta-Review · Area_Chair_kcve · 2024-12-22

**Metareview:**

This work is a survey/meta-analysis of more than 50 recent papers on scaling laws in deep learning; it identifies key methodological decisions involved in scaling law research (functional form, model training, data, etc) and explains inconsistencies in the literature through this lens. The work also conducts its own experiments to explore how significantly conclusions can vary depending on experimental design choices.

Scaling laws is a very active research area, yet it has not reached consensus on methodological best-practices: different papers make slightly different choices of how to measure & fit laws, and these choices can sometimes affect the conclusions. Moreover, as noted by this work, many papers omit crucial experimental details which impact scaling laws. It is thus nontrivial for researchers to understand the current status of scaling laws by reading the literature, especially for those who are not aware of the “folklore”.
Reviewers appreciated the work this paper does to clarify the field; some quotes from reviewers:
* “It is of great value for the community to "recap" scaling law results and provide a critical discussion” (Reviewer kn4j)
* “I very much believe there is merit in a survey or pointing out these problems; as it stands, however, the paper is foremost “just” a survey, and I am not convinced this merits publishing at the conference.” (Reviewer vpGv)

In the reviewer discussion, the main concern preventing a higher score was the fact that this paper is primarily a survey, and it is unclear whether ICLR accepts such works.
There is no official policy on whether surveys are in-scope for ICLR.
However, historically ICLR has been welcoming to papers that are valuable and of interest to the community, even if they are outside the traditional scope of ML conferences (e.g. the einops paper, ICLR 2022 Oral: https://openreview.net/forum?id=oapKSVM2bcj ).
In this specific case, having read both the paper and the review discussion, I recommend acceptance for the following reasons:
* It is clear that the work is valuable for the community (especially newcomers), and clarifies the state of the active but not-yet-mature research field of scaling laws.
*  The work is not purely a survey; it includes aspects of meta-analysis wherein it identifies sources of inconsistencies in prior work. This goes beyond simply surveying known results.
* I believe this type of work should be encouraged at ICLR: deep learning is a fast-moving field, and there should be incentives to occasionally reflect on recent progress and coalesce results into a collective understanding.

**Additional Comments On Reviewer Discussion:**

See above

---

### Decision · Program_Chairs · 2025-01-22

Accept (Poster)